# WAVTOKENIZER: AN EFFICIENT ACOUSTIC DISCRETE CODEC TOKENIZER FOR AUDIO LANGUAGE MODELING

**Shengpeng Ji** ♠♡∗ **Ziyue Jiang** ♠∗ **Wen Wang** ♡ **Yifu Chen** ♠ **Minghui Fang** ♠
**Jialong Zuo** ♠ **Qian Yang** ♠ **Xize Cheng** ♠ **Zehan Wang** ♠ **Ruiqi Li** ♠ **Ziang Zhang** ♠
**Xiaoda Yang** ♠ **Rongjie Huang** ♣ **Yidi Jiang** ♡ **Qian Chen** ♡ **Siqi Zheng** ♡ **Zhou Zhao** ♠†

♠Zhejiang University & ♡Alibaba Group & ♣Fundamental AI Research (FAIR), Meta

Code and Checkpoint: `https://github.com/jishengpeng/WavTokenizer`

## ABSTRACT

Language models have been effectively applied to modeling natural signals, such as images, video, speech, and audio. A crucial component of these models is the tokenizer, which compresses high-dimensional natural signals into lower-dimensional discrete tokens. In this paper, we introduce **WavTokenizer**, which offers several advantages over previous state-of-the-art (SOTA) acoustic codec models in the audio domain: 1) **extreme compression.** By compressing the layers of quantizers and the temporal dimension of the discrete codec, one-second audio of **24kHz** sampling rate requires only a single quantizer with 40 or 75 tokens. 2) **improved subjective reconstruction quality.** Despite the reduced number of tokens, WavTokenizer achieves SOTA reconstruction quality with outstanding UTMOS scores and **also inherently contains richer semantic information**. Specifically, we achieve these results by designing a broader VQ space, extending contextual windows, improving attention networks, and introducing a powerful multi-scale discriminator and an inverse Fourier transform structure. We conduct extensive reconstruction experiments in the domains of speech, audio, and music. WavTokenizer exhibits competitive to superior performance across various objective and subjective metrics compared to SOTA models. We also evaluate WavTokenizer on semantic representation, VQ utilization, and adaptability to generative models. Comprehensive ablation studies confirm the necessity of each module in WavTokenizer.

## 1 INTRODUCTION

Recently, significant achievements have been made by large language models (LLMs) (Brown et al., 2020) in audio generative tasks, including multiple-speaker speech syntheses (Wang et al., 2023; Kharitonov et al., 2023; Jiang et al., 2023; 2024; Ji et al., 2024c), music generation (Agostinelli et al., 2023), and audio generation (Kreuk et al., 2022). Furthermore, the integration of the speech modality into unified large multimodal models also has garnered significant attention, such as SpeechGPT (Zhang et al., 2023a), AnyGPT (Zhan et al., 2024), GPT-4o, and Moshi (Défossez et al., 2024). These successes can be largely attributed to the utilization of discrete acoustic codec representations produced by neural codec models (Zeghidour et al., 2021; Défossez et al., 2022; Kumar et al., 2023). These discrete acoustic codec models bridge the gap between continuous speech signal and discrete-token-based language models, by discretizing high-rate audio signals into a finite set of tokens, hence enabling the application of LLM architectures to audio data.

Most end-to-end discrete codec models (Défossez et al., 2022; Wu et al., 2023) adopt a three-stage structure consisting of an encoder, a Residual Vector Quantization (RVQ) module, and a decoder. The encoder performs downsampling of the audio signal in the time domain to obtain compressed audio frames. Each compressed audio frame is then quantized by a series of quantizers, with each quantizer

---

∗Equal contribution.
†Corresponding author.

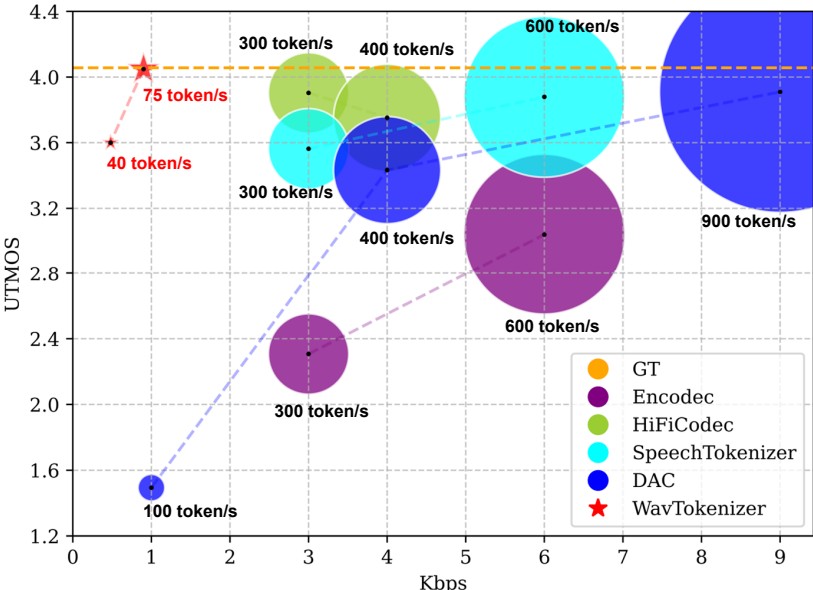

Figure 1: Comparison between different acoustic codec models. The y-axis UTMOS reflects reconstruction quality (UTMOS highly correlates with human evaluations), the x-axis kbps represents audio compression levels. The size of circles represents the number of discrete tokens per second.

operating on the residual of the previous one. The number of quantizers determines the overall bitrate. The decoder, on the other hand, performs upsampling in the time domain to reconstruct the audio signal from the quantizer outputs. Existing acoustic codec models (Kumar et al., 2023) demonstrate impressive reconstruction quality, and generative models based on discrete codecs are now capable of synthesizing speech at near-human levels. However, two important directions are worth exploring beyond the current acoustic codec models, namely, *high bitrate compression* and *semantic richness*.

**Higher Bitrate Compression.** The compression level of current codec models still warrants exploration to achieve higher compression. Two aspects merit optimization: the number of quantizers and the temporal dimension of the codec. While some efforts have reduced the quantity of quantizers from eight to four (Yang et al., 2023; Ji et al., 2024a), we argue that **a single quantizer layer fundamentally differs from multiple quantizers**. When the number of quantizers exceeds one, downstream models require additional design efforts, such as VALL-E's (Wang et al., 2023) AR and NAR structures, SoundStorm's (Borsos et al., 2023; Ji et al., 2024b) parallel generation, Music-Gen's (Copet et al., 2024; Peng et al., 2024) slanted autoregressive structure, and UniAudio's (Yang et al., a) global and local attention structures. Conversely, with a single quantizer, speech modalities can be directly autoregressively embedded into large multimodal models (Touvron et al., 2023). Additionally, high temporal dimensions of codecs, such as DAC's (Kumar et al., 2023) requirement of 900 tokens per second, substantially degrade language model generation quality and increase resource consumption.

**Richer Semantic Information.** Considering the gap between the codec's reconstruction paradigm and the generative paradigm of downstream models (SpeechTeam, 2024; Chu et al., 2024), incorporating more semantic information in codec can facilitate weakly supervised text-to-speech generation. Moreover, many large multimodal models (Chu et al., 2023; Tang et al., 2023) adopt the Whisper paradigm for understanding tasks (the target for generation is acoustic codec); hence, incorporating additional semantic information into acoustic codec models could help unify the understanding and generation processes in multimodal models. While many approaches attempt to introduce semantic information through distillation, additional pre-trained semantic modules (Zhang et al., 2023b) can interfere with the unified modeling of music and audio. Moreover, this distillation-based strategy may limit the potential of codec models. Exploring more elegant approaches to *directly* integrate semantic information into the codec remains an open question.

In this paper, we introduce **WavTokenizer**, a discrete acoustic codec model capable of reconstructing 24kHz speech, music, and audio using only 40 or 75 tokens per second. WavTokenizer achieves high reconstruction quality with extreme compression while enhancing the semantic richness of the codec. Specifically, WavTokenizer enhances audio reconstruction quality by employing a multi-scale discriminator and an inverse Fourier transform upsampling structure from the vocoder in the decoder. To compress the codec from multiple quantizers to a single one, we discover that *expanding the VQ space*, alongside employing recent *K-means clustering initialization* and *random awakening strategies*, can significantly compress audio representations while maintaining high codebook utilization. Additionally, expanding the contextual window for speech modeling and incorporating attention networks in the decoder not only balance reconstruction quality and compression but also enrich the semantic information. Our contributions can be summarized as follows:

- **Conceptual Contributions.** We introduce the concept of compressing the quantizer layers of acoustic codec models to a single quantizer for the first time and enhancing semantic information of the codec without disrupting the codec paradigm for modeling music and audio. Inspired by a detailed analysis of the codebook space in Section 3.2, we propose aligning the **large speech space with the textual vocabulary** and show the potential of **large speech space** as a latent form of a unique language.
- **Methodological Contributions.** Utilizing K-means clustering initialization and random awakening strategies on the VQ codebook space, we design an expanded VQ space for compressing the codec model to a single quantizer. Furthermore, we design extended contextual modeling windows and add attention mechanisms in the decoder. The integration of an inverse Fourier transform module and multi-scale discriminator in the vocoder also contributes to improved reconstruction.
- **Experimental Contributions.** WavTokenizer surpasses the current state-of-the-art (SOTA) models' subjective reconstruction performance on speech, music, and audio, with only 75 tokens per second. It achieves comparable results with 40 or 75 tokens per second across broader metrics. Additional experiments demonstrate the superiority of WavTokenizer over competitive baseline models regarding semantic information, codebook utilization, and performance in generative models. Rigorous ablation studies confirm the necessity of each component in WavTokenizer. We will open-source the entire codebase and pretrained models for WavTokenizer.

## 2 RELATED WORK

In recent times, neural acoustic codecs (Zeghidour et al., 2021; Défossez et al., 2022; Kumar et al., 2023) have demonstrated remarkable capabilities in reconstructing high-quality audio at low bitrates. Typically, these methods employ an encoder to extract deep features in a latent space, which are subsequently quantized before being fed into the decoder. Given that acoustic tokens, compared to semantic tokens, can support audio, speech, and music domains, and their rich acoustic details can eliminate the need for cascading architectures in downstream generative models (Kharitonov et al., 2023; Huang et al., 2024b) or large multimodal models (SpeechTeam, 2024; Anastassiou et al., 2024), current optimization directions for acoustic codec models can be categorized as follows:

**Pursuing Better Reconstruction Quality.** AudioDec (Wu et al., 2023) demonstrates the importance of discriminators. PromptCodec (Pan et al., 2024) enhances representation capabilities through additional input prompts. DAC (Kumar et al., 2023) significantly improves reconstruction quality through techniques like quantizer dropout and a multi-scale Short-Time Fourier Transform (STFT) based discriminator. Vocos (Siuzdak, 2023) eliminates codec noise artifacts using a pre-trained Encodec with an inverse Fourier transform vocoder. HILCodec (Ahn et al., 2024) introduces the MFBD discriminator to guide codec modeling. APCodec (Ahn et al., 2024) further enhances reconstruction quality by incorporating ConvNextV2 modules in the encoder and decoder.

**Enhancing Compression.** HiFi-Codec (Yang et al., 2023) proposes a parallel GRVQ structure and achieves good speech reconstruction quality with just four quantizers. Language-Codec (Ji et al., 2024a) introduces the MCRVQ mechanism to evenly distribute information across the first quantizer and only requires four quantizers to achieve excellent performance across various generative models. Regarding achieving high compression with a single quantizer, Single-Codec (Li et al., 2024) is most related to our work. Throughout the progression of our work on WavTokenizer, Single-Codec designs additional BLSTM, hybrid sampling, and resampling modules to ensure basic performance with a single quantizer; however, different from the impressive reconstruction performance of our WavTokenizer, Single-Codec's reconstruction quality is uncompetitve (with UTMOS only 3.0).

**Deepening Understanding of the Codec Space.** TiCodec (Ren et al., 2024) models the codec space by distinguishing between time-independent and time-dependent information. FACodec (Ju et al.) decouples the codec space into content, style, and acoustic-detail modules. Additionally, recognizing the importance of semantic information in generative models, recent efforts start integrating semantic information into codec models. RepCodec (Huang et al., 2024c) learns a vector quantization codebook by reconstructing speech representations from speech encoders like HuBERT (Hsu et al., 2021) or Data2vec (Baevski et al., 2022). SpeechTokenizer (Zhang et al., 2023b) enriches the semantic content of the first quantizer through semantic distillation. FunCodec (Du et al., 2023) makes semantic tokens optional and explores different combinations. SemanticCodec (Liu et al., 2024) uses quantized semantic tokens and reconstructs acoustic information using an audio encoder and diffusion model. Although the semantic codecs achieve good audio reconstruction quality, they disrupt the encoder-VQ-decoder paradigm of acoustic codec models and introduce additional training costs.

**Compared to the aforementioned approaches, WavTokenizer achieves impressive reconstruction results with only one quantizer and through 40 or 75 tokens**. In contrast, for one second of speech, DAC (Kumar et al., 2023) requires 900 tokens, with 9 quantizers. **Furthermore, WavTokenizer explores enhancing semantic information by strengthening the capabilities of the Codec itself.**

## 3 WAVTOKENIZER

Our model is built on the framework of VQ-GANs, following the same paradigm as Sound-Stream (Zeghidour et al., 2021) and EnCodec (Défossez et al., 2022). Specifically, WavTokenizer passes the raw audio $X$ through three modules: 1) A full convolution encoder network that takes the input audio and generates a latent feature representation $Z$; 2) **A single quantizer** discretizes $Z$ to generate a discrete representation $Z_q$. 3) **An improved decoder** that reconstructs the audio signal $\tilde{X}$ from the compressed latent representation $Z_q$. The model is trained *end-to-end*, optimizing a reconstruction loss applied over both time and frequency domains, along with a perceptual loss in the form of discriminators operating at different resolutions.

Considering that WavTokenizer is designed as a discrete token representation for large audio language models, the focus should be on the subjective reconstruction quality of the codec (audio fidelity) and semantic content information. In Figure 1, we visualize the relationship between bitrates and UTMOS metrics (Saeki et al., 2022) across different codec models. As shown in the Figure 1, WavTokenizer achieves SOTA reconstruction quality with only 75 tokens. Notably, WavTokenizer facilitates extreme compression bitrates and achieves a fair UTMOS score of 3.6 at 0.48 kpbs.

### 3.1 ENCODER

Following Encodec (Défossez et al., 2022), the encoder consists of a 1D convolution with $C$ channels and a kernel size of 7 followed by $B$ convolution blocks. Each convolution block is composed of a single residual unit followed by a downsampling layer consisting of a strided convolution, with a kernel size twice of the stride $S$. The residual unit contains two convolutions with kernel size of 3 and a skip-connection. The number of channels is doubled whenever downsampling occurs. The convolution blocks are followed by a two-layer LSTM for sequence modeling and a final 1D convolution layer with a kernel size of 7 and $D$ output channels. Following Encodec, we set $C = 32$, $B = 4$, $D = 512$, and use ELU (Clevert et al., 2015) as a non- linear activation function. For the stride $S$, we employ two configurations, (2, 4, 5, 8) and (4, 5, 5, 6), to ensure that WavTokenizer can downsample 24 kHz speech by factors of 320 and 600 along the time dimension.

### 3.2 RETHINKING THE VECTOR QUANTIZATION SPACE

WavTokenizer aims to compress speech representations into the codebook space of a single quantizer. This allows for the seamless serialization of speech and elimination of the need of hierarchical design in downstream models across channel dimensions (Wang et al., 2023; Yang et al., a; Borsos et al., 2023). Initially, we attempt to rely solely on a single quantizer for reconstruction during training, without changing any structure; however, we find the results suboptimal. **Considering the vast vocabulary space in natural language, we hypothesize that treating speech as a unique language might yield better results**. Motivated by this hypothesis, we conduct the following analysis. First, we expand the codebook space from $2^{10}$ to $2^{14}$. Next, we train on 585 hours of LibriTTS and then

visualize the probability distribution of codebooks on the LibriTTS **test-clean** dataset, as shown in Figure 2 (a). We observe a concentration of the speech vocabulary to the left of $2^{12}$, indicating the potential merit of utilizing the larger $2^{12}$ speech vocabulary space since the current codec codebooks $2^{10}$ may not fully represent the speech space. More analyses in Appendix D verify that increasing the training dataset size does not lead to higher codebook space utilization and the codebook space trained with *4000 hours multilingual data* remains concentrated to the left of $2^{12}$.

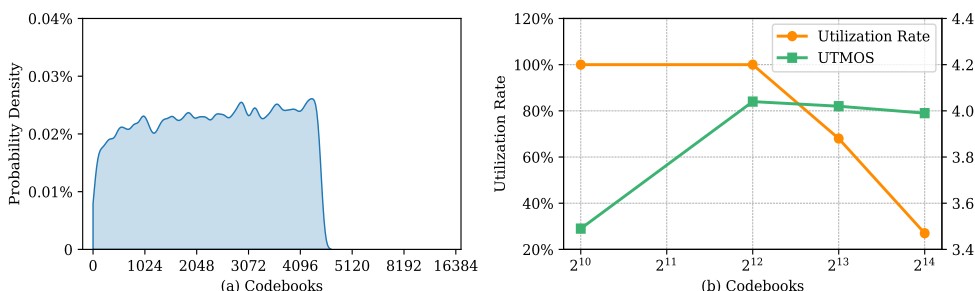

Figure 2: The visualization analysis of WavTokenizer's quantized codebook space. Figure (a) illustrates the probability distribution of each codebook index (1-16384) on the LibriTTS test-clean across different codebook spaces. Figure (b) examines the relationship between reconstruction quality in terms of UTMOS and codebook utilization rate across different codebook spaces.

Simply expanding the quantized codebook space could lead to lower utilization rates. To mitigate this issue, leveraging recent advancements in codec models (Défossez et al., 2022; Ju et al.), we use k-means clustering to initialize the codebook vectors. We adjust the number of cluster centers to 200 to align with the larger codebook space. During training, for each input, the code is selected, assigned, and updated using an exponential moving average with a decay of 0.99, and codes unassigned for several batches are replaced with input vectors randomly sampled from the current batch. This forced activation strategy (Dhariwal et al., 2020) helps ensure effective utilization of the large codebook space. We analyze the relationship between codebook utilization rate and reconstruction result, after applying the aforementioned k-means clustering initialization and random awakening strategies. Figure 2 (b) confirms that expanding the corresponding codebook space appropriately can reduce information loss caused by compressing the hierarchical RVQ structure into a single quantizer. Speech can be effectively reconstructed under a serialized quantizer structure, with a codebook space of $2^{12}$ achieving a favorable balance between codebook utilization and reconstruction quality. Furthermore, the experiments in Section 4.3 explore the potential of a large codebook space in discrete acoustic codecs as a specialized form of language representations.

### 3.3 IMPROVED DECODER

As in FACodec (Ju et al.), we believe that the decoder plays a more crucial role than the encoder during the acoustic codec reconstruction process. Upsampling and reconstructing audio from the highly compressed information in WavTokenizer is particularly challenging. Notably, WavTokenizer does not employ a mirrored decoder upsampling structure (Kumar et al., 2023), a standard practice that uses a stack of dilated convolutions to increase the receptive field, and uses transposed convolutions to sequentially upsample the feature sequence to the waveform. Since this standard design is known to be susceptible to aliasing artifacts, following Vocos (Siuzdak, 2023), we maintain consistent feature resolution at all depths and achieve waveform upsampling through inverse Fourier transform. In the decoder, the target audio signal $\tilde{X}$ is represented using STFT:

$$STFT(\tilde{X}_{[m,k]}) = \sum_{n=0}^{N} \tilde{X}[n]\, w\,[n-m]\, e^{-j2\pi kn/K} \tag{1}$$

where $K$ denotes the number of frequency points after performing Discrete Fourier Transform (DFT), $k$ denotes the frequency index, $N$ denotes the number of points in the sampled sequence, with $n$ denoting a particular sample point, and $m$ denoting the index length. In the practical implementation, STFT is performed by applying a series of Fast Fourier Transforms (FFTs) to overlapping and windowed frames of data. The window function advances or hops through time to create these frames.

Moreover, to *directly* enhance the semantic modeling capability of the acoustic codec model, rather than adding various semantic tokens (Zhang et al., 2023b), we hypothesize that incorporating an attention network module (Rombach et al., 2022) into the decoder may enhance information reconstruction and semantic modeling. Although attention models have proven their scalability and high performance in broader tasks (Dosovitskiy et al., 2020; Fang et al., 2024; Huang et al., 2024a; Yang et al., b), the encoder-decoder structure in the acoustic codec model remains fully convolutional. Concerns may arise about the potential extrapolation issues of attention models when modeling long sequences during inference, given that acoustic codec models often train on randomly selected *short* one-second audio clips. However, our experiments show that WavTokenizer achieves good reconstruction even for long audio sequences during inference. Additionally, we find that simply expanding contextual modeling windows to three seconds for WavTokenizer with attention modules could further improve codec reconstruction during training and in turn the reconstruction quality during inference. This is probably because one-second clips, including silence, may contain insufficient semantic information; hence, increasing the contextual modeling window size helps the Codec model better capture context. We validate these findings through detailed ablation studies (Section 4.3). We investigate various configurations of introducing attention modules to the encoder, the decoder, or both, and find that adding the attention module to the decoder only is beneficial, and adding it before the ConvNext module (Liu et al., 2022) appears to be optimal.

Therefore, for the representation of the intermediate signals $Z_q$ after quantization, WavTokenizer only needs to input $Z_q$ into the conv1D layer, attention block, ConvNeXt blocks, which serves as the fundamental backbone. Subsequently, a Fourier transform is performed on the real-valued signals. Following Vocos (Siuzdak, 2023), the ConvNeXt Block first embeds the input features into a hidden dimensionality and then applies a sequence of convolutional blocks. Each block is composed of a large-kernel-sized depthwise convolution, followed by an inverted bottleneck that projects features into a higher dimensionality using pointwise convolution. GELU activations (Hendrycks & Gimpel, 2016) are used within the bottleneck, and Layer Normalization is employed between the blocks. Regarding the transformation of real-valued signals, we utilize a single side band spectrum, resulting in $n_{fft}/2 + 1$ coefficients per frame. Since we parameterize the model to output both phase and magnitude values, the activations of the hidden dimensions are projected into a tensor $h$ with $n_{fft} + 2$ channels. Finally, the inverse Fourier transform $\mathcal{F}^{-1}$ is used to directly reconstruct the final audio.

### 3.4 THE ADVANCED DISCRIMINATOR AND THE LOSS FUNCTIONS

We use the adversarial loss to promote perceptual quality. Following Vocos (Siuzdak, 2023), We employ the open-source multi-period discriminator (MPD) (Kong et al., 2020), the single band amplitude only and multi band complex multi-resolution discriminator (MRD) (Jang et al., 2021). Furthermore, to learn discriminative features about a specific sub-band and provide a stronger gradient signal to the generator, following (Kumar et al., 2023), we use a complex STFT discriminator (Zeghidour et al., 2021) at multiple time-scales (Défossez et al., 2022). We adopt a hinge loss formulation instead of the least squares GAN objective, as suggested by (Zeghidour et al., 2021). The discriminator training loss $\mathcal{L}_{dis}(X, \tilde{X})$ is as follows:

$$\frac{1}{K} \sum_{k=1}^{K} max(0, 1 - D_k(X)) + max(0, 1 + D_k(\tilde{X})) \tag{2}$$

$K$ denotes the number of discriminators, $D_k$ denotes the $k$-th discriminator. The training loss for the generator of WavTokenizer consists of four components: quantizer loss, mel-spectrum reconstruction loss, adversarial loss, and feature matching loss. The quantizer loss is defined as follows:

$$\mathcal{L}_q(Z, Z_q) = \sum_{i=1}^{N} \left\| Z_i - \hat{Z}_i \right\|_2^2 \tag{3}$$

The mel-spectrum reconstruction loss is defined as follows:

$$\mathcal{L}_{mel}(X, \tilde{X}) = \left\| Mel(X) - Mel(\tilde{X}) \right\|_1 \tag{4}$$

Furthermore, we define the adversarial loss as a hinge loss over the logits of these discriminators:

$$\mathcal{L}_{adv} = \frac{1}{K} \sum_{k=1}^{K} max(0, 1 - D_k(\tilde{X})) \tag{5}$$

The feature matching loss, denoted as $\mathcal{L}_{feat}$, is calculated as the mean of the distances between the $l$th feature maps of the $k$th subdistriminator:

$$\mathcal{L}_{feat} = \frac{1}{K * L} \sum_k \sum_l \left\| D_k^l(X) - D_k^l(\tilde{X}) \right\|_1 \tag{6}$$

The total training loss of the generator, $\mathcal{L}_{gen}$, is computed as:

$$\mathcal{L}_{gen} = \lambda_q \mathcal{L}_q + \lambda_{mel} \mathcal{L}_{mel} + \lambda_{adv} \mathcal{L}_{adv} + \lambda_{feat} \mathcal{L}_{feat} \tag{7}$$

where $\lambda_q, \lambda_{mel}, \lambda_{adv}, \lambda_{feat}$ are hyper-parameters.

## 4 EXPERIMENTS

### 4.1 EXPERIMENTAL SETUP

**Dataset.** Due to constrained computation resources, we conduct the training process only on a subset of common publicly available datasets. WavTokenizer is trained on approximately 8K hours of data. For the speech domain, we use LibriTTS (Zen et al., 2019), VCTK (Veaux et al., 2016), and a subset of CommonVoice (Ardila et al., 2019)(3000 hours are randomly selected). For the audio domain, we utilize a subset of AudioSet (Gemmeke et al., 2017)(2000 hours are randomly selected); and for the music domain, we employ the Jamendo (Bogdanov et al., 2019) and MusicDB (Rafii et al., 2017) datasets. We evaluate speech reconstruction performance of codec in clean and noisy environments using the LibriTTS *test-clean* and *test-other* sets respectively, and assess audio and music reconstruction performance using the AudioSet eval and MusicDB test sets respectively. On the other hand, for most confirmatory experiments, such as the ablation experiments, we evaluate the results with the WavTokenizer trained only on LibriTTS.

**Baselines.** We select the state-of-the-art (SOTA) codec models as the baselines for WavTokenizer. To ensure fair comparisons, we employ the official weights provided by Encodec (Défossez et al., 2022)[1], HiFi-Codec (Yang et al., 2023)[2], Vocos (Siuzdak, 2023)[3], SpeechTokenizer (Zhang et al., 2023b)[4], and DAC (Kumar et al., 2023)[5] frameworks.

**Evaluation metrics and Implementation Details.** For **objective** evaluation of discrete codec models, following Vocos (Siuzdak, 2023), we employ the UTMOS (Saeki et al., 2022) automatic Mean Opinion Score (MOS) prediction system. UTMOS can yield scores highly correlated with human evaluations, closer to human perception than PESQ (Perceptual Evaluation of Speech Quality) (Rix et al., 2001), but it is restricted to 16 kHz sample rate. we also adopt the metrics in speech enhancement fields, such as PESQ, STOI (Short-time Objective Intelligibility), and the F1 score for voiced/unvoiced classification (V/UV F1). In addition to these objective metrics, following Encodec (Défossez et al., 2022), we also employ the **subjective** MUSHRA evaluation to assess the reconstruction performance of the codec. We employ the common subjective CMOS evaluation metrics to assess the performance of the downstream TTS model with the codec models. Details of the subjective evaluations are in Appendix C. **Training and Inference Settings** are detailed in Appendix A.

### 4.2 MAIN RESULTS

**Evaluation on Reconstruction.** We compare the *speech* reconstruction performance of WavTokenizer with a broad selection of SOTA and competitive codec models as baselines on LibriTTS *test-clean* (4837 samples), LibriTTS *test-other* (5120 samples), and LJSpeech (13100 samples), which correspond to audio reconstruction in clean, noisy, and out-of-domain environments, respectively. Notably, RVQ-based codec models often select quantizers with varying bandwidths during training. To ensure fair comparisons, we use the quantizers that the baseline models are trained on. The results are shown in Table 1. We observe the following: 1) **WavTokenizer achieves impressive results on the UTMOS metric, with WavTokenizer at 0.9 kbps surpassing the current SOTA DAC model at 9 kbps on all test sets**. Since the UTMOS metric closely aligns with human perception

---

[1]https://github.com/facebookresearch/encodec
[2]https://github.com/yangdongchao/AcademiCodec
[3]https://github.com/gemelo-ai/vocos
[4]https://github.com/ZhangXInFD/SpeechTokenizer
[5]https://github.com/descriptinc/descript-audio-codec

Table 1: **Objective reconstruction results** of different codec models on LibriTTS *test-clean* (clean environment), LibriTTS *test-other* (noisy environment), and *LJSpeech dataset* (out-of-domain environment). **Nq** denotes the **n**umber of **q**uantizers. **GT** denotes ground truth waveforms. Best results from models with a single quantizer (hence directly comparable to WavTokenizer) are in bold.

| Dataset | Model | Bandwidth ↓ | Nq ↓ | token/s ↓ | UTMOS ↑ | PESQ ↑ | STOI ↑ | V/UV F1 ↑ |
|---|---|---|---|---|---|---|---|---|
| | GT | - | - | - | 4.0562 | - | - | - |
| | DAC | 9.0kpbs | 9 | 900 | 3.9097 | 3.9082 | 0.9699 | 0.9781 |
| | Encodec | 6.0kbps | 8 | 600 | 3.0399 | 2.7202 | 0.9391 | 0.9527 |
| | Vocos | 6.0kbps | 8 | 600 | 3.6954 | 2.8069 | 0.9426 | 0.9437 |
| | SpeechTokenizer | 6.0kbps | 8 | 600 | 3.8794 | 2.6121 | 0.9165 | 0.9495 |
| | DAC | 4.0kbps | 4 | 400 | 3.4329 | 2.7378 | 0.9280 | 0.9572 |
| LibriTTS *test-clean* | HiFi-Codec | 3.0kbps | 4 | 400 | 3.7529 | 2.9611 | 0.9405 | 0.9617 |
| | HiFi-Codec | 4.0kbps | 4 | 300 | 3.9035 | 3.0116 | 0.9446 | 0.9576 |
| | Encodec | 3.0kbps | 4 | 300 | 2.3070 | 2.0517 | 0.9007 | 0.9198 |
| | Vocos | 3.0kbps | 4 | 300 | 3.5390 | 2.4026 | 0.9231 | 0.9358 |
| | SpeechTokenizer | 3.0kbps | 4 | 300 | 3.5632 | 1.9311 | 0.8778 | 0.9273 |
| | DAC | 1.0kbps | 1 | 100 | 1.4940 | 1.2464 | 0.7706 | 0.7941 |
| | WavTokenizer | 0.5kbps | 1 | 40 | 3.6016 | 1.7027 | 0.8615 | 0.9173 |
| | WavTokenizer | 0.9kbps | 1 | 75 | **4.0486** | **2.3730** | **0.9139** | **0.9382** |
| | GT | - | - | - | 3.4831 | - | - | - |
| | DAC | 9.0kpbs | 9 | 900 | 3.3566 | 3.7595 | 0.9576 | 0.9696 |
| | Encodec | 6.0kbps | 8 | 600 | 2.6568 | 2.6818 | 0.9241 | 0.9338 |
| | Vocos | 6.0kbps | 8 | 600 | 3.1956 | 2.5590 | 0.9209 | 0.9202 |
| | SpeechTokenizer | 6.0kbps | 8 | 600 | 3.2851 | 2.3269 | 0.8811 | 0.9205 |
| | DAC | 4.0kbps | 4 | 400 | 2.9448 | 2.5948 | 0.9083 | 0.9404 |
| LibriTTS *test-other* | HiFi-Codec | 4.0kbps | 4 | 400 | 3.0750 | 2.5536 | 0.9126 | 0.9387 |
| | HiFi-Codec | 3.0kbps | 4 | 300 | 3.3034 | 2.6083 | 0.9166 | 0.9318 |
| | Encodec | 3.0kbps | 4 | 300 | 2.0883 | 2.0520 | 0.8835 | 0.8926 |
| | Vocos | 3.0kbps | 4 | 300 | 3.0558 | 2.1933 | 0.8967 | 0.9051 |
| | SpeechTokenizer | 3.0kbps | 4 | 300 | 3.0183 | 1.7373 | 0.8371 | 0.8907 |
| | DAC | 1.0kbps | 1 | 100 | 1.4986 | 1.2454 | 0.7505 | 0.7775 |
| | WavTokenizer | 0.5kbps | 1 | 40 | 3.0545 | 1.6622 | 0.8336 | 0.8953 |
| | WavTokenizer | 0.9kbps | 1 | 75 | **3.4312** | **2.2614** | **0.8907** | **0.9172** |
| | GT | - | - | - | 4.3794 | - | - | - |
| | DAC | 9.0kpbs | 9 | 900 | 4.3007 | 3.9022 | 0.9733 | 0.9757 |
| | Encodec | 6.0kbps | 8 | 600 | 3.2286 | 2.6633 | 0.9441 | 0.9555 |
| | Vocos | 6.0kbps | 8 | 600 | 4.0332 | 2.9258 | 0.9497 | 0.9459 |
| | SpeechTokenizer | 6.0kbps | 8 | 600 | 4.2373 | 2.6413 | 0.9316 | 0.9452 |
| | DAC | 4.0kbps | 4 | 400 | 3.8109 | 2.7616 | 0.9338 | 0.9524 |
| *LJSpeech* | HiFi-Codec | 4.0kbps | 4 | 400 | 4.1656 | 2.7629 | 0.9446 | 0.9497 |
| | HiFi-Codec | 3.0kbps | 4 | 300 | 4.2692 | 2.9091 | 0.9485 | 0.9469 |
| | Encodec | 3.0kbps | 4 | 300 | 2.3905 | 2.0194 | 0.9058 | 0.9326 |
| | Vocos | 3.0kbps | 4 | 300 | 3.7880 | 2.5006 | 0.9310 | 0.9388 |
| | SpeechTokenizer | 3.0kbps | 4 | 300 | 3.9908 | 2.0458 | 0.9021 | 0.9299 |
| | DAC | 1.0kbps | 1 | 100 | 1.4438 | 1.2084 | 0.7822 | 0.8095 |
| | WavTokenizer | 0.5kbps | 1 | 40 | 4.0186 | 2.1142 | 0.9093 | **0.9406** |
| | WavTokenizer | 0.9kbps | 1 | 75 | **4.2580** | **2.4923** | **0.9312** | 0.9397 |

of audio quality (Saeki et al., 2022), these results validate that **WavTokenizer maintains excellent reconstruction quality under extreme compression**. 2) When compared to the SOTA DAC model with a *single* quantizer (hence directly comparable to WavTokenizer, shown in the shaded regions of Table 1), WavTokenizer with 40 and 75 tokens remarkably outperforms DAC with 100 tokens across all metrics. **To the best of our knowledge, WavTokenizer is the first codec model capable of effectively reconstructing audio with a single quantizer**. 3) On objective metrics STOI, PESQ, and F1 score, WavTokenizer also performs comparably to the Vocos model with four quantizers and the SpeechTokenizer model with eight quantizers. 4) In noisy environments and out-of-domain scenarios, WavTokenizer demonstrates strong robustness and generalizability across all metrics.

We further evaluate the reconstruction performance of the codec models on the broader range of *music* and *audio* domains. Following Encodec, we used MUSHRA as the metric for subjective evaluation. As shown in Table 2, WavTokenizer at 0.9 kbps outperforms the SOTA DAC model at 9 kbps in reconstruction quality on the speech, music, and audio domains. These results further demonstrate that **WavTokenizer is capable of maintaining high subjective reconstruction performance on speech, music, and audio, with an extremely limited number of tokens.**

**Evaluation on Semantic Representation.** We evaluate the semantic richness of different codec models on the ARCH benchmark (La Quatra et al., 2024). Notably, we opt not to utilize the

conventional Superb benchmark (Yang et al., 2021) due to its exclusive focus on the speech domain, while ARCH enables further assessment of a Codec model in music and audio realms. The ARCH benchmark comprises 12 datasets in speech, music, audio domains (details in Appendix B). We extract embeddings corresponding to the discrete codebooks of an acoustic codec model as its respective representations and evaluate the classification accuracy of the codec model on ARCH datasets using its representations. For fair comparisons, we evaluate Encodec and DAC models on semantic representation, as they are under the same paradigm as WavTokenizer. The experimental results, as shown in Table 3, demonstrate that WavTokenizer substantially outperforms DAC and Encodec configured with a single quantizer or two quantizers on classification accuracy. Remarkably, on the AM and SLURP datasets in the speech domain, the MTT and IRMAS datasets in the music domain, and the FSD50K and VIVAE datasets in the audio domain, **WavTokenizer surpasses DAC with nine quantizers and Encodec with eight quantizers on classification performance**.

Table 2: The **subjective reconstruction results** using MUSHRA (comparative scoring of samples) of codec models on speech, music and audio domains. **Nq** denotes the **n**umber of **q**uantizers.

| Model | Bandwidth ↓ | Nq ↓ | token/s ↓ | *LibriTTS test-clean* ↑ | *MusicDB* ↑ | *Audioset* ↑ |
|---|---|---|---|---|---|---|
| GT | - | - | - | 96.4±1.2 | 95.3±1.7 | 95.8±2.1 |
| DAC | 9.0kpbs | 9 | 900 | 92.8±1.8 | 92.6±2.4 | 92.7±1.5 |
| Encodec | 6.0kbps | 8 | 600 | 78.6±1.9 | 76.9±1.6 | 81.2±1.8 |
| DAC | 1.0kbps | 1 | 100 | 58.4±2.4 | 57.6±2.1 | 56.8±1.4 |
| WavTokenizer | 0.9kbps | 1 | 75 | **96.1±2.3** | **92.9±2.2** | **94.4±1.6** |

Table 3: The **semantic representation (speech, music, audio)** evaluation of different codec models on ARCH Benchmark in terms of **classification accuracy**. **Nq** represents the **n**umber of **q**uantizers.

| Model | Nq ↓ | token/s ↓ | RAVDESS ↑ | SLURP ↑ | EMOVO ↑ | AM ↑ | FMA ↑ | MTT ↑ | IRMAS ↑ | MS-DB ↑ | ESC50 ↑ | US8K ↑ | FSD50K ↑ | VIVAE ↑ |
|---|---|---|---|---|---|---|---|---|---|---|---|---|---|---|
| DAC | 9 | 900 | 0.3750 | 0.0779 | 0.2363 | 0.6926 | 0.3504 | 0.2805 | 0.4023 | 0.6014 | 0.2594 | 0.4032 | 0.1297 | 0.3440 |
| Encodec | 8 | 600 | 0.2881 | 0.0636 | 0.2261 | 0.4388 | 0.2790 | 0.1993 | 0.3671 | 0.3917 | 0.1925 | 0.3055 | 0.1091 | 0.3005 |
| DAC | 4 | 400 | 0.3194 | 0.0782 | 0.2346 | 0.6838 | 0.3379 | 0.2784 | 0.3833 | 0.5942 | 0.2580 | 0.3824 | 0.1293 | 0.3342 |
| Encodec | 4 | 300 | 0.2951 | 0.0660 | 0.2193 | 0.4301 | 0.2728 | 0.1934 | 0.3684 | 0.3656 | 0.1790 | 0.3097 | 0.1099 | 0.2710 |
| Encodec | 2 | 150 | 0.2743 | 0.0627 | 0.2193 | 0.3649 | 0.2816 | 0.1900 | 0.3704 | 0.3245 | 0.1699 | 0.2960 | 0.1065 | 0.2630 |
| DAC | 1 | 100 | 0.2500 | 0.0713 | 0.2278 | 0.6287 | 0.3304 | 0.2502 | 0.3572 | 0.5137 | 0.2065 | 0.3350 | 0.1295 | 0.2991 |
| WavTokenizer | 1 | 75 | **0.3255** | **0.0802** | **0.3163** | **0.6957** | **0.3417** | **0.2835** | **0.4117** | **0.5764** | **0.2550** | **0.3975** | **0.1392** | **0.3563** |

**Evaluation on Downstream Generative Tasks.** We evaluate WavTokenizer's performance on downstream generative tasks exemplified by **text-to-speech synthesis (TTS)** (Ren et al., 2020). We adopt an autoregressive language model backbone. Specifically, we follow the MusicGen paradigm (Copet et al., 2024), which expands the acoustic codec sequence through autoregressive prediction, and modify the open-sourced ParlerTTS model (Lyth & King, 2024) accordingly. We use the ParlerTTS 600M model configuration and its default hyperparameters, and train TTS models based on DAC and WavTokenizer representations on the LibriTTS dataset. Each model is trained for 40 epochs on 8 A800 80GB GPUs.

The results are shown in Table 4. In terms of audio quality (CMOS-Q) and audio prosody (CMOS-P), the speech synthesis model trained on WavTokenizer's single-layer quantizer representations outperforms that trained on DAC's 9-layer quantizer representations. This demonstrates that: 1) **Acoustic models with a single-layer quantizer show significant potential in downstream autoregressive audio generation models**, and 2) **Speech synthesis models using large-codebook acoustic representations can also synthesize high-quality audio**, suggesting that large-codebook acoustic spaces have the potential to model speech as a special form of text. We will validate these findings on large multimodal models (Zhan et al., 2024) based on acoustic codec models that are *trained on larger datasets* in future work.

Table 4: The **Subjective Evaluations** of various acoustic codec models for **downstream text-to-speech synthesis models** on the LibriTTS test set. **GT** denotes ground truth waveforms.

| Model | Bandwidth ↓ | Nq ↓ | CMOS-Q↑ | CMOS-P↑ |
|---|---|---|---|---|
| GT | - | - | 0.22 | 0.26 |
| DAC | 9.0kbps | 9 | -0.35 | -0.29 |
| WavTokenizer | 0.9kbps | 1 | **0.00** | **0.00** |

Table 5: Impact of codebook scale. Utilization rate reflects codebook's usage efficiency.

| Model | Codebooks | Utilization rate | UTMOS ↑ | PESQ ↑ | STOI ↑ |
|---|---|---|---|---|---|
| WavTokenizer | 16384 | 27% | 3.9989 | 2.3600 | 0.8129 |
| WavTokenizer | 8192 | 68% | 4.0220 | 2.3916 | 0.9156 |
| WavTokenizer | 4096 | 100% | 4.0486 | 2.3730 | 0.9139 |
| WavTokenizer | 1024 | 100% | 3.4967 | 1.7781 | 0.8660 |

Table 6: Impact of the contextual modeling window size.

| Model | Codebooks | windows | UTMOS ↑ | PESQ ↑ | STOI ↑ |
|---|---|---|---|---|---|
| WavTokenizer | 4096 | 1 | 3.7448 | 2.0112 | 0.8944 |
| WavTokenizer | 4096 | 3 | 4.0486 | 2.3730 | 0.9139 |
| WavTokenizer | 4096 | 5 | 4.0448 | 2.3556 | 0.9127 |

## 4.3 ABLATION STUDY

Due to limited compute resources, we use 585 hours LibriTTS training data for training WavTokenizer and conduct ablation studies on reconstruction performance on the LibriTTS ***test-clean subset.*** For all ablation experiments, we use WavTokenizer with a single quantizer operating at 0.9 kbps.

***Codebook size.*** we evaluate varying codebook sizes on the performance of WavTokenizer. We record the frequency of each codebook entry on LibriTTS ***test-clean***. As shown in Table 5 and discussed in Section 3.2, we observe significant potential for expansion in the typical codebook space, even under extreme compression with a single quantizer, when combined with advanced training strategies (Section 3.2). We find that expanding the codebook size from the typical 1024 to 4096 significantly enhances audio quality, with the UTMOS gain by 0.55, PESQ gain by 0.6, and STOI gain by 0.5. However, excessively large codebook spaces (16384) can lead to reduced codeboook utilization.

***Contextual window size.*** Most Codec models are trained on randomly selected one-second audio clips. With an attention module in WavTokenizer's decoder, as shown in Table 6, using a three-second contextual window further enhances the reconstruction quality. We hypothesize that a one-second window may contain insufficient information and be more affected by silence. Longer contextual window may enhance attention module's ability to capture relevant semantics.

***Multi-scale STFT discriminator.*** As shown in Table 7, the multi-scale STFT discriminator (MSTFTD) enhances the reconstruction quality, albeit increasing the training time. The improvement is probably due to the fact that MSTFTD splits STFT into sub-bands and learns discriminative features for a specific sub-band, hence providing a stronger gradient signal to WavTokenizer's generator, in turn improving high-frequency prediction and mitigating aliasing artifacts.

***Attention module.*** We remove the attention module from WavTokenizer's decoder, resulting in *WavTokenizer w/o attention*. As shown in Table 7, removing the attention module degrades WavTokenizer's performance more than MSTFTD. Ablation results on impact of the attention module on semantic representation are in Appendix E.

***Improved decoder.*** We replace the improved decoder based on the inverse Fourier transform with an up-sampling structure mirroring the encoder, denoted as *WavTokenizer w/ mirror decoder*. As shown in Table 7, a purely mirrored structure substantially compromises the reconstruction performance of WavTokenizer under extreme compression. This underscores the importance of a robust decoder in ensuring the reconstruction performance of codec models under high compression.

Table 7: Ablation on the multi-scale STFT discriminator (MSTFTD), the atention module, and switching from our improved decoder to a mirror decoder, in WavTokenizer.

| Model | UTMOS ↑ | PESQ ↑ | STOI ↑ | V/UV F1 ↑ |
|---|---|---|---|---|
| WavTokenizer | **4.0486** | **2.3730** | **0.9139** | **0.9382** |
| w/ mirror decoder | 2.7782 | 1.5007 | 0.8249 | 0.8820 |
| w/o attention module | 3.6020 | 1.9332 | 0.8734 | 0.9067 |
| w/o MSTFTD | 3.7806 | 2.1270 | 0.9008 | 0.9269 |

## 5 CONCLUSION

In this paper, we introduce WavTokenizer, a codec model capable of quantizing one second speech, music and general audio into 75 or 40 tokens with a single quantizer. Compared to SOTA acoustic codec models, WavTokenizer maintains high subjective reconstruction quality and preserves rich semantic information even under extreme compression. The limitation of WavTokenizer and future work are discussed in Appendix F.

## ACKNOWLEDGMENTS

This work was done at Alibaba Group during the internship of Shengpeng Ji. I would like to express my sincere gratitude to my internship mentors, Qian Chen, Siqi Zheng, and Wen Wang, for their valuable guidance and support. This work was supported by the National Natural Science Foundation of China under Grant No.62222211.

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

## A    TRAINING AND INFERENCE SETTINGS

We train WavTokenizer up to 2 million iterations, with 1 million iterations allocated to training the generator and the discriminator respectively, on 8 NVIDIA A800 80G GPUs. Throughout the entire training process, all input speech, music, and audio samples are resampled to 24 kHz, and the batch size is 40. We uniformly truncate excessively long segments in the training data to a fixed length of 10 seconds and subsequently perform a random crop of the waveform to obtain audio snippets of 3-second duration to be fed into WavTokenizer. WavTokenizer is optimized using the AdamW optimizer with an initial learning rate of 2e-4 and betas set to (0.9, 0.999). The learning rate is decayed based on a cosine schedule.

## B    THE ARCH BENCHMARK

The ARCH benchmark comprises twelve datasets within the speech, music, audio domain. Emotional Speech and Song (RAVDESS) (Livingstone & Russo, 2012), Audio-MNIST (AM) (Becker et al., 2024), Spoken Language Understanding Resource Package (SLURP) (Bastianelli et al., 2020), and EMOVO dataset (Costantini et al., 2014) assess performance in the Speech domain. ESC-50 (Piczak, 2015), US8K (Salamon et al., 2014), FSD50K (Fonseca et al., 2021), and VIVAE (Holz et al., 2022) assess performance on Acoustic Events. FMA (Defferrard et al., 2016), MTT (Law et al., 2009), IRMAS (Bosch et al., 2012), and MS-DB (Rafii et al., 2017) assess performance in the Music domain.

## C    SUBJECTIVE EVALUATIONS

For the subjective evaluations, we follow the MUSHRA protocol (Series, 2014), using both a hidden reference and a low anchor. Annotators are recruited using a crowd-sourcing platform, in which they are asked to rate the perceptual quality of the provided samples in a range between 1 to 100. We randomly select 50 samples from each category of the test set and ensure at least 10 annotations per sample. To filter out noisy annotations and outliers, we remove annotators who rate the reference recordings less then 90 in at least 20% of the cases, or rate the low-anchor recording above 80 more than 50% of the time.

For CMOS-Q and CMOS-P evaluations, we randomly choose 40 samples from the LibriTTS *test-set* for the subjective evaluation, and each audio is listened to by at least 10 testers. We analyze the CMOS in two aspects: CMOS-Q (quality, clarity and high-frequency details), CMOS-P (Speech rate, pauses, and pitch). We instruct the testers to focus on the aspect in question and ignore the other aspect when scoring the aspect being considered.

## D    ABLATION EXPERIMENTS ON MORE TRAINING DATA AND CODEBOOK SPACE

We further evaluate whether a larger training dataset would elevate the upper bound of the codebook space. The results, shown in Table 8, present codebook utilization on the LibriTTS *test-clean* dataset. We find that increasing the training dataset size from 585 hours to 4000 hours does not lead to higher codebook space utilization. Moreover, through visualizing the probability distribution, we observe that the codebook space trained with larger datasets remains concentrated on the left side of the 4096 range.

Table 8: The ablation study investigates the impact of dataset size on codebook utilization.

| Model | Dataset | Codebooks | Utilization rate | UTMOS ↑ | PESQ ↑ | STOI ↑ |
|---|---|---|---|---|---|---|
| WavTokenizer | 585 Hours | 16384 | 27% | 3.9989 | 2.3600 | 0.8129 |
| WavTokenizer | 4000 Hours | 16384 | 26.5% | 3.9465 | 2.3721 | 0.8217 |

## E  ABLATION ON THE ATTENTION MODULE AND THE CONTEXTUAL MODEL SIZE ON THE ARCH BENCHMARK

We conduct ablation study on the impact of the attention module and the extended context modeling window in WavTokenizer on semantic information, with experiments performed on the ARCH (La Quatra et al., 2024) speech-domain datasets. The experimental results, as shown in Table 9, indicate that adding an attention module to the decoder and also extending the context window in the codec model improve the preservation of semantic information within the codec.

Table 9: The ablation study of the attention module and the contextual window size on the semantic information in WavTokenizer.

| Model | Nq ↓ | token/s ↓ | RAVDESS ↑ | SLURP ↑ | EMOVO ↑ | AM ↑ |
|---|---|---|---|---|---|---|
| WavTokenizer w/o attention w/o extended windows | 1 | 75 | 0.2614 | 0.0643 | 0.2368 | 0.6192 |
| WavTokenizer | 1 | 75 | **0.3255** | **0.0802** | **0.3163** | **0.6957** |

## F  LIMITATION AND FUTURE WORK

While WavTokenizer is capable of reconstructing high-quality audio using only 75 tokens and demonstrates the potential of a single-layer quantizer with a large codebook space in downstream generative models, current acoustic codec models lack the understanding capabilities (ASR) found in semantic models (Hsu et al., 2021). This limitation constrains the development of codec models within unified multimodal understanding and generation frameworks, such as the GPT-4o paradigm. At present, distillation methods used in acoustic codec models serve more as temporary solutions. Unlike the powerful decoder we design in WavTokenizer, in future work, we aim to explore a more robust encoder module that can further improve compression, reconstruction, and semantic information retention. Specifically, a key focus of our future work on WavTokenizer will be to explore **elegant methods** for significantly enhancing the semantic capacity of discrete tokens within the encoder.

Additionally, we will train WavTokenizer on hundreds of thousands of hours of speech data, and will verify whether acoustic codec models with single-layer quantizers and large codebook spaces, trained on a large amount of speech data, can truly align speech as a special form of language to the text space within unified large multimodal models.

## G  WAVTOKENIZER AT 16KHZ AND 48KHZ SAMPLING RATE

We have augmented the reconstruction evaluations by training both 16kHz and 48kHz versions of WavTokenizer using LibriTTS while maintaining a consistent downsampling factor of 320x. The hyperparameters are kept at their default settings. Experimental results on the LibriTTS test-clean dataset (4,837 samples) are presented in Table 10.

We find that WavTokenizer delivers high-quality and comparable audio reconstruction performance across different sampling rates of 16kHz, 24kHz, and 48kHz. Notably, at 16kHz sampling rate, WavTokenizer can effectively reconstruct audio with high fidelity using only 50 tokens. We also observed that all objective metrics, except for UTMOS, exhibit an increasing trend of increasing scores when the number of tokens increases. In contrast, UTMOS for WavTokenizer remains relatively stable. We hypothesize that the reconstruction quality of WavTokenizer should be stable across different sampling rates, to human perception of audio quality; hence, this observation of stable UTMOS aligns with prior finding Saeki et al. (2022) that UTMOS highly correlates with human subjective auditory perception.

## H  RECONSTRUCTION SPEED

We evaluate the reconstruction speed of Semanticodec Encodec, DAC, and WavTokenizer on a single NVIDIA A100 80G GPU on the LibriTTS *test-clean* dataset. We calculate the real-time factor (RTF) by dividing the total reconstruction time by the duration of the generated audio. The results are shown

Table 10: **Objective reconstruction performance** for WavTokenizer at different sampling rates on Librispeech test-clean. **Sr** denotes sampling rate. The results of WacTokenizer at 24kHz are compared to baseline models in Table 1.

| Methods | Sr | Tokens/s | UTMOS ↑ | PESQ ↑ | STOI ↑ | V/UV F1 ↑ |
|---|---|---|---|---|---|---|
| WavTokenizer | 16000 | 50 | 3.9606 | 2.3240 | 0.9095 | 0.9375 |
| WavTokenizer | 24000 | 75 | 4.0486 | 2.3730 | 0.9139 | 0.9382 |
| WavTokenizer | 48000 | 150 | 3.9582 | 2.7230 | 0.9350 | 0.9496 |

in Table 11. Notably, despite incorporating the attention module in the decoder of WavTokenizer, its reconstruction speed remains remarkably fast. This can be attributed to two factors: (1) the use of a fast inverse Fourier transform, following Vocos, and (2) the low bitrate of WavTokenizer. These results demonstrate the high reconstruction efficiency of WavTokenizer.

Table 11: Reconstruction speed (measured by RTF) of different codec models on reconstruction on the LibriTTS *test-clean* dataset. **RTF** is computed by dividing the total reconstruction time by the duration of the generated audio.

| Methods | RTF ↓ |
|---|---|
| SemantiCodec | 0.9483 |
| Encodec | 0.0175 |
| DAC | 0.0144 |
| WavTokenizer | 0.0098 |

# I   MORE TTS RESULTS

Note that in Section 4.2, we present the subjective evaluation results of TTS models in the acoustic codec and LLM architecture in Table 4. We further supplement these results with synthesis accuracy and speaker similarity results. Building on the pre-trained TTS model using WavTokenizer as presented in Section 4.2, we further evaluate the synthesis accuracy, measured by WER, and speaker similarity SPK, computed by using WavLM to extract speaker embeddings for cosine similarity, on the **zero-shot TTS task**, following the same settings as the VALL-E-continue version (Wang et al., 2023). As shown in Table 12, in the audio codec and autoregressive LLM architecture for audio generation, with an identical generative model, **the TTS model trained on WavTokenizer (0.9kbps) yields substantially lower WER than the TTS model trained on SOTA acoustic codec model DAC (9.0kbps), as 5.1% versus 6.9%, and also achieves better speaker similarity performance, as 0.61 versus DAC's 0.59**. This further confirms that WavTokenizer demonstrates excellent performance in the downstream TTS tasks.

Table 12: Word Error Rate (WER) and Speaker Similarity (SPK) of various acoustic codec models for **downstream speech synthesis models**. **GT** denotes ground truth waveforms. To evaluate Spk between the original prompt and the synthesized speech, we utilize the base-plus-sv version of WavLM.

| Model | Bandwidth ↓ | Nq ↓ | WER ↓ | SPK ↑ |
|---|---|---|---|---|
| GT | - | - | 2.4 | - |
| DAC | 9.0kbps | 9 | 6.9 | 0.59 |
| WavTokenizer | 0.9kbps | 1 | **5.1** | **0.61** |

