# OpenReview forum: "WavTokenizer: an Efficient Acoustic Discrete Codec Tokenizer for Audio Language Modeling"
_ICLR.cc/2025/Conference — ICLR 2025 Poster_

### Official Review · Reviewer_h5oK · 2024-10-31

**Soundness:** 3
**Presentation:** 2
**Contribution:** 2
**Rating:** 5
**Confidence:** 5

**Summary:**

This paper introduces WavTokenizer, a GAN-based neural audio codec that uses a single vector quantization (VQ) codebook, in contrast to previous methods that rely on multiple residual vector quantization (RVQ) codebooks. This results in a bitrate as low as 480 bits per second, and to recover quality, the authors propose replacing the time-domain decoder with a Fourier-based decoder, preceded by attention layers. The ablation study confirms these design choices, and a comprehensive evaluation, including both subjective and objective metrics, shows that WavTokenizer maintains competitive reconstruction quality with state-of-the-art models.

**Strengths:**

- The paper effectively motivates and addresses an important problem in neural audio codecs - quantizing audio into a single sequence of tokens rather than multiple sequences, which complicates modeling for downstream tasks.
- It presents useful findings regarding decoder design, which are supported by the ablation study. In particular, a Fourier-based decoder combined with an attention layer yields better results, while a time-domain decoder with attention performs surprisingly worse.

**Weaknesses:**

While the motivation for scaling the single VQ codebook to many entries is clear, the paper falls short of achieving high codebook utilization when expanding beyond a size of 4096. What the authors list as contributions in VQ, such as k-means initialization and random restarts, are in fact well-established techniques in neural audio compression, and this paper doesn’t offer any novel methods to improve codebook usage. This is somewhat disappointing, given that a key focus of the paper is to provide a single quantizer. More experimentation to scale the VQ codebook is needed, as the current contribution feels more incremental and may be better suited for a different venue.

**Questions:**

1. Did the authors try other techniques to improve VQ codebook utilization? The current approach closely mirrors EnCodec, but DAC demonstrates that low-dimensional code lookups and L2-normalization can significantly improve RVQ scalability. A useful reference to consider is [1], which shows effective scaling strategies in image reconstruction that could be also valuable for audio codecs. I recommend continuing work on the paper, as a single-quantizer audio codec has potential, but the paper’s current scientific contribution is limited.


2. What is the motivation for evaluating semantic representation? Neural audio codecs are expected to encode low-level acoustic features rather than abstract semantic concepts. Comparing audio codecs on semantic representation could be misleading, especially given the results e.g. all codecs score below 10% on the SLURP dataset, while self-supervised models like HuBERT achieve nearly 50% (not shown in this paper). This gap calls into question the statement on line 445 that "WavTokenizer effectively captures rich semantic information" which may be an overclaim. That said, audio codecs might have a significant impact on representation learning, as shown by EnCodecMAE [2], so it may be more appropriate to treat semantic representation as a downstream task. WavTokenizer's single codebook could be particularly useful for discrete targets in BERT-like setups.

[1] Zhu, Lei, et al. "Scaling the Codebook Size of VQGAN to 100,000 with a Utilization Rate of 99%."

[2] Pepino, Leonardo, Pablo Riera, and Luciana Ferrer. "EnCodecMAE: Leveraging neural codecs for universal audio representation learning."

---

> ### Author Response · Authors · 2024-11-21
> **Response to Reviewer h5oK (Part 1/2)**
>
> Thank you very much for your time and effort in reviewing our paper and your valuable feedback. We hope our response below fully resolves your concerns and questions.
>
> **[Regarding Weakness and Question 1 Improve VQ Codebook Utilization]**
>
> **1.1.** Your summary of VQ is highly insightful. Please note that **WavTokenizer's primary contribution in VQ design lies in its analysis and design choices to expand the codebook space under extremely low kbps conditions.**  Improving codebook utilization efficiency is not the focus of our innovations in the original paper.
>
> During the rebuttal phase, we supplemented our work with experiments exploring the impact of low-dimensional code lookups and L2-normalization, which are used in DAC, on codebook utilization rate and reconstruction performance.  However, during training, we observed that the gradient of the commit loss was not stable. As shown in the table below, using low-dimensional code lookups and L2-normalization in WavTokenizer did not result in significant gains in codebook utilization rate. In fact, under identical configurations, WavTokenizer experienced a slight drop in reconstruction performance with low-dimensional code lookups and L2-normalization.
>
>
> | Methods |  Codebooks | Utilization rate |  UTMOS ↑ | PESQ ↑ | STOI ↑ |
> | :------------: | :------------: | :------------: | :------------: | :------------: | :------------: |
> | WavTokenizer | 8192 | 68% | 4.0220 | 2.3916 | 0.9156 |
> | WavTokenizer | 4096 | 100% | 4.0486 | 2.3730 | 0.9139 |
> | WavTokenizer w/ low-dimensional, L2-normalization | 8192 | 71% | 3.9683 | 2.2678 | 0.9044 |
> | WavTokenizer w/ low-dimensional, L2-normalization | 4096 | 100% | 3.9615 | 2.2423 | 0.9036 |
>
> **1.2.** Additionally, you mentioned VQGAN-LC, which is indeed an excellent work. VQGAN-LC initializes the codebook with features extracted from a pre-trained CLIP model, creating a well-structured latent space that better aligns with the encoder's output distribution. However, the latent space defined by an external pre-trained model may limit the model's generalization capability across diverse datasets and might restrict its applicability to the speech domain.
>
> It is worth noting that the latest VQ work, which surpasses VQGAN-LC across various image-related tasks, has also reported results in the speech domain. The speech-related findings in some other VQ works show that even with state-of-the-art VQ techniques, there was no significant improvement in reconstruction quality for speech tasks. Therefore, as both the latest VQ work and your comments suggest, we believe that WavTokenizer offers valuable inspiration to the community by encouraging further exploration of codebook expansion along the lines of WavTokenizer.
>
>
> **1.3.** In the speech domain, in prior works including DAC, innovations in VQ predominantly involved applying existing techniques and validating them through experiments. The goal has generally been to apply advanced VQ techniques for building effective acoustic codec models. We find that after extensive training (16 * A100 * 60 days), WavTokenizer demonstrates strong subjective reconstruction performance and also compatibility with large language models (LLMs), as shown in the downstream TTS results in Table 4 and Table 16, which is one of the reasons for the high rating of 10 from Reviewer 3XJA and rating of 8 from Reviewer wU96 on our paper. We hope these explanations are useful for your reassessment of our contributions.

---

> ### Author Response · Authors · 2024-11-21
> **Response to Reviewer h5oK (Part 2/2)**
>
> **[Regarding Question 2 Motivation for evaluating semantic representation]**
>
>
> **2.1.** The statement "Comparing audio codecs based on semantic representation could be misleading" holds true in the context of traditional audio codecs, where evaluating acoustic information is often of greater importance. However, a current trend involves integrating semantic information into acoustic codecs, as seen in models like SpeechTokenizer, FunCodec, XCodec, and Mimi. We believe that this integration serves a dual purpose: **it allows models like GPT-4o and Moshi to better leverage textual knowledge of pre-trained text LLMs and incorporate speech modality without diminishing the text LLM's capabilities. Embedding semantic information into audio codecs facilitates better alignment with textual modality.** Moreover, in multimodal large models (e.g., mini-Omni1 and 2, Qwen-Audio1 and 2), it is common to use strong semantic tokens like Whisper for audio comprehension tasks and employ acoustic tokens for audio generation tasks. Consequently, **incorporating semantic information into audio codecs will likely help unify the understanding and generation processes.** Therefore, we reported evaluations of semantic representation of WavTokenizer and various audio acoustic codecs in the part of Evaluation on Semantic Representation in Section 4.2.
>
> **2.2.** As you rightly pointed out, in Table 3, we only demonstrate that WavTokenizer, at a lower bitrate of 75 tokens/s, is capable of achieving semantic information comparable to the 9-layer SOTA acoustic codec DAC model (900 tokens/s) and outperforms DAC also configured with a single quantizer and in similar low bitrates as WavTokenizer, which is why we referred to it as "rich semantic information". We sincerely apologize for any confusion caused by the term "rich". We have updated the part of Evaluation on Semantic Representation in Section 4.2 to clarify this point.
>
> We expect WavTokenizer to underperform semantic tokens such as HuBERT, EncodecMAE and codecs such as SpeechTokenizer that distill from semantic tokens, on semantic representation benchmarks such as ARCH. Our experimental results align with this expectation: on the ARCH benchmark, WavTokenizer underperforms semantic tokens like HuBERT, EncodecMAE, and in some cases, it falls short compared to SpeechTokenizer. To address this limitation, as explained in the original paper (Line 887), we plan to significantly enhance the semantic information in WavTokenizer by strengthening the encoder. We appreciate your understanding on this matter.
>
> **2.3.** It is also worth noting that one of our contributions about semantic in this paper is demonstrating that **the low-bitrate WavTokenizer retains comparable semantic information to the SOTA 9-layer DAC**. We achieved this performance through **a more elegant way** of directly enhancing the semantic modeling capability of the acoustic codec model itself, rather than adding various semantic tokens like SpeechTokenizer and Mimi. Specifically, WavTokenizer achieves this performance by incorporating an attention network module into the decoder and simply expanding contextual modeling window. Please find the ablation study in Line 864-877 of the original paper. We believe the research direction we explored is important because prior works based on distilling semantic tokens into audio codec models may compromise the unified modeling of acoustic codecs for music and audio and potentially impose limitations on the performance of the acoustic codecs. To improve semantic information of WavTokenizer, as explained in Appendix F Limitation and Future work of the original paper, a key focus of our future work on WavTokenizer will be to explore the line of elegant methods for significantly enhancing the semantic capacity of discrete tokens within the encoder.
>
> Finally, we would like to express our gratitude again for your time and effort in reviewing our paper. Please do not hesitate to let us know if you have any further concerns or comments. We would be happy to address them.

---

> > ### Comment · Reviewer_h5oK · 2024-11-26
> >
> > ### VQ
> >
> > > "Please note that WavTokenizer's primary contribution in VQ design lies in its analysis and design choices to expand the codebook space under extremely low kbps conditions. Improving codebook utilization efficiency is not the focus of our innovations in the original paper."
> >
> > Could you please clarify your contribution in VQ? In Section 1 you list k-means clustering initialization and random awakening as your methodological contributions. But these methods are well-established and were previously proposed in SoundStream (Zeghidour et al., 2021) and Jukebox (Dhariwal et al., 2020).
> >
> > If your contribution is to increase the codebook size, how can "improving codebook utilization efficiency" not be a focus? You essentialy have to address the problem of low codebook utilization (that you observe in your experiments) to effectively increase codebook size. Based on the presented results, it seems this was not successful.
> >
> > That said, the idea of a neural codec with a single quantizer is appealing and I personally like the direction you are exploring. However, to provide competitive performance to RVQ (especially for higher sampling rates and diverse acoustic conditions), the codebook needs to grow by an order of magnitude, not merely two times, as proposed in this paper.
> >
> > ### Semantic representation
> >
> > "Embedding semantic information into audio codecs facilitates better alignment with textual modality."
> > I agree with this. But as you pointed out in the rebuttal, WavTokenizer underperforms against semantic baselines and you did not include this comparison in the paper. While you present results showing a roughly 2% improvement in semantic representation over DAC, you omit that it performs far worse than HuBERT. This raises doubt about presenting the method as an improvement in semantic representation.
> >
> > Given the limited scientific contribution and unresolved concerns, I'm keeping my rating.

---

### Official Review · Reviewer_wU96 · 2024-11-01

**Soundness:** 3
**Presentation:** 3
**Contribution:** 4
**Rating:** 8
**Confidence:** 5

**Summary:**

This paper introduces WavTokenizer, while preserving the classical acoustic codec model paradigm, achieves high-quality audio reconstruction using only 40 or 75 tokens per second. By proposing a larger codebook space, integrating attention mechanisms, and extending the context window, WavTokenizer demonstrates impressive results in audio reconstruction, semantic understanding, and downstream TTS tasks.

**Strengths:**

- The acoustic codec representation model is a crucial technology in the current speech domain. WavTokenizer addresses one of the core challenges in the field by achieving high-quality audio reconstruction with only 40 or 75 tokens.
- WavTokenizer introduces a novel single-layer quantizer concept, demonstrating its potential in TTS tasks and offering a promising single-layer solution for codec-LLM architectures.
- From a methodological standpoint, WavTokenizer revisits vector quantization (VQ) in the speech domain and proposes a larger codebook space, a more powerful decoder (with attention mechanisms), and an extended context modeling window. These innovations appear to be effective.
- The model achieves strong experimental results across reconstruction tasks, semantic understanding tasks, and downstream TTS tasks.
- The open-sourcing of the complete training and inference code, along with model weights, will contribute to the development of the research community.

**Weaknesses:**

Overall, this work does not present significant weaknesses. However, the design of a highly powerful decoder in WavTokenizer raises some concerns. Specifically, I am concerned that the increased model parameters and the introduction of attention mechanisms may potentially slow down the codec's reconstruction speed?

**Questions:**

- How was VQ utilization calculated in Figure 2(b) of the paper?
- It seems that WavTokenizer is not inherently streamable. Can WavTokenizer be extended to support streaming encoding and decoding?

---

> ### Author Response · Authors · 2024-11-21
> **Response to Reviewer wU96**
>
> Thanks for your valuable feedback, and we hope our response fully resolves your concerns.
>
>
> **[Regarding Question 1 Inference Speed]**
>
> We evaluated the reconstruction speed of Semanticodec Encodec, DAC, and WavTokenizer on a single NVIDIA A100 80G GPU. By dividing the total reconstruction time by the duration of the generated audio, we calculated the real-time factor (RTF). The results are shown in the table below. Notably, despite incorporating the attention module in the decoder of WavTokenizer, its reconstruction speed remains remarkably fast. This can be attributed to two factors: (1) The use of a fast inverse Fourier transform in the decoder, following Vocos; and (2) the low bitrate of WavTokenizer. These results demonstrate the high reconstruction efficiency of WavTokenizer. We have updated the paper with this comparison in Appendix I.
>
> | Methods |  RTF↓ |
> | :------------: | :------------: |
> | Semanticodec| 0.9483 |
> | Encodec | 0.0175 |
> | DAC     | 0.0144 |
> | WavTokenizer | 0.0098 |
>
> **[Regarding Question 2]**
>
> As discussed in Section 3.2 of the original paper, we evaluated the VQ utilization on the LibriTTS test-clean dataset, consisting of 4,837 audio samples. First, we used WavTokenizer to obtain discrete tokens for each audio sample. Then, based on the codebook size, we calculated the frequency of occurrence of each discrete token. In Figure 2(a), we present the probability density distribution of token frequencies. In Figure 2(b), we define a token as "utilized" if it occurs more than once. Using this method, we further computed the utilization rate.
>
> **[Regarding Question 3]**
>
> Since most acoustic codec models are non-causal, our paper also includes results for non-causal reconstruction. Similar to other acoustic codec models, WavTokenizer supports streaming reconstruction. This can be achieved by replacing convolutional modules with causal convolutions (e.g., adding padding to the left). Importantly, as WavTokenizer incorporates an attention module, it is necessary to implement a streaming version of causal attention by constructing a mask matrix. Notably, Mimi employs a similar causal attention mechanism, achieving excellent performance. Therefore, a streaming WavTokenizer can also be implemented using this approach.

---

> > ### Comment · Reviewer_wU96 · 2024-11-21
> >
> > Thanks for your responses. The replies answer my concerns well. I intend to keep my score unchanged.

---

### Official Review · Reviewer_VpBu · 2024-11-04

**Soundness:** 2
**Presentation:** 3
**Contribution:** 1
**Rating:** 3
**Confidence:** 5

**Summary:**

The paper introduces WavTokenizer, a single-vq codec aimed at simplifying and improving the current audio language modeling approaches by replacing codec with multiple-vq. It claims to achieve competitive reconstruction quality while enhancing integration with language models.

**Strengths:**

The overall approach is sound.  The paper is straightforward and easy to understand, with a clear structure that makes it accessible to readers. Additionally, the introduction provides a good overview of the context and background, helping readers to understand the problem.

**Weaknesses:**

The paper introduces WavTokenizer, a single-VQ codec designed to replace the current codec + LLM audio generation systems. However, there is no experimental evidence to support that WavTokenizer can effectively achieve this. Existing codec + LLM systems can generally be categorized into RVQ-style generation (e.g., VALL-E, MusicGen, Moshi) or semantic-to-acoustic approaches (e.g., SpearTTS, SeedTTS, MusicLM), all of which demonstrate strong performance.

Firstly, there are no experiments showing that WavTokenizer combined with an LLM outperforms any of the existing systems in practical applications such as TTS or music generation. Secondly, when considering the codec itself, WavTokenizer's reconstruction quality is significantly worse than RVQ-based codecs, and its semantic performance falls short compared to semantic token like HuBERT or WavLM.

Given these shortcomings, it is difficult to see any tangible improvements or significant contributions that WavTokenizer offers to the current field of audio generation.

**Questions:**

1, Missing baselines: SemantiCodec[1], Single-Codec[2], Mimi (Moshi) [3].

2, Weak Semantic Performance: Compared to the ARCH benchmark (https://huggingface.co/spaces/ALM/ARCH), the semantic performance of WavTokenizer is far worse than models like HuBERT base. Therefore, the claim of "rich semantic information" does not hold up well. Moreover, it is unclear why WavTokenizer's semantic capabilities were compared only against acoustic codec models, rather than semantic codecs. Why not compare against semantic codecs like SpeechTokenizer[4], SemantiCodec[1], or Mini[3]?

3, Single-VQ Assumption: The entire premise of the paper is based on the assumption that using a single-VQ codec is better for audio language modeling compared to RVQ. But is this correct? The paper mentions that RVQ-based codecs like DAC require 900 tokens per second, which is true, but it fails to acknowledge that RVQ codecs typically have a temporal resolution of only 50Hz, and the most advanced models like Mini have even lower resolution at 12.5Hz. In practice, language models typically need to model at frequencies of 50Hz or less. So, what is the real advantage of a single-VQ codec with 40 or 75 tokens per second? Moshi and Mini have already demonstrated success in low-latency speech dialogue applications, but what about single-VQ? Is reconstructing all acoustic details truly beneficial for language modeling?  Does an LLM truly need to model timbre and detailed acoustic features, or is focusing solely on semantic content sufficient?  Therefore, you must at least demonstrate the effectiveness of single-VQ in practical applications, as stated in the weakness section, rather than constructing the paper solely based on this assumption.

[1]: Liu H, Xu X, Yuan Y, et al. SemantiCodec: An Ultra Low Bitrate Semantic Audio Codec for General Sound[J]. arXiv preprint arXiv:2405.00233, 2024.

[2]: Li H, Xue L, Guo H, et al. Single-Codec: Single-Codebook Speech Codec towards High-Performance Speech Generation[J]. arXiv preprint arXiv:2406.07422, 2024.

[3]:Défossez A, Mazaré L, Orsini M, et al. Moshi: a speech-text foundation model for real-time dialogue[J].

[4]:Zhang X, Zhang D, Li S, et al. Speechtokenizer: Unified speech tokenizer for speech large language models[J]. arXiv preprint arXiv:2308.16692, 2023.

---

> ### Author Response · Authors · 2024-11-21
> **Response to Reviewer VpBu (Part 1/4)**
>
> Thank you very much for your time and effort in reviewing our paper and your valuable feedback. We hope our response below fully resolves your concerns and questions.
>
> **[Regarding Weakness 1 and Question 2: Lack codec+LLM audio generation experiments, reconstruction quality worse than RVQ-based codecs, semantic performance worse than semantic token like HuBERT or WavLM]**
>
>
> We sincerely appreciate your valuable suggestions. However, we believe there may be some misunderstandings or you missed the results already presented in the main body of the original paper, which we will clarify below.
>
> **1.1 TTS experiments:** As presented **in Table 4 and discussed in Line 464-480 of the original paper, we have already included text-to-speech synthesis (TTS) results based on the codec+LLM audio generation framework.** As described in Line 464-480, we use the open-sourced ParlerTTS 600M model configuration and its default hyperparameters, and train TTS models based on the SOTA acoustic codec model DAC and WavTokenizer representations on the LibriTTS dataset. Line 472-474 summarized the results in Table 4. **In terms of audio quality (CMOS-Q) and audio prosody (CMOS-P), the speech synthesis model trained on WavTokenizer’s single-layer quantizer representations (0.9kbps) outperforms the TTS model trained on DAC’s 9-layer quantizer representations (9.0kbps)**, as 0.00 CMOS-Q and 0.00 CMOS-P against -0.35 CMOS-Q and -0.29 CMOS-P from DAC. Note that CMOS-Q and CMOS-P for ground truth waveforms are 0.22 and 0.26 respectively.
>
>
> After submission, building on the pre-trained TTS model, we further supplemented these results by evaluating the synthesis accuracy, measured by WER, and speaker similarity SPK, computed by using WavLM to extract speaker embeddings between the original prompt and the synthesized speech for computing cosine similarity,  on the zero-shot TTS task, following the same settings as the VALL-E-continue version. As shown in Appendix K in the revised paper, **the updated TTS results show that in the codec + autoregressive LLM architecture, the TTS model trained on WavTokenizer (0.9kbps) yields substantially lower WER than the TTS model trained on SOTA acoustic codec model DAC (9.0kbps), as 5.1% versus 6.9%, and also achieves better speaker similarity**, as 0.61 versus DAC's 0.59. Moreover, similar TTS results using WavTokenizer have been independently reported in other works by other teams, such as OuteTTS and Lina-Speech.
>
>
>
> | Methods |  Bandwidths↓ |  Nq ↓ | WER ↓ | SPK↑ |
> | :------------: | :------------: | :------------: | :------------: |  :------------: |
> | GT | - | - | 2.4 | - |
> | DAC | 9.0kbps | 9 | 6.9 | 0.59 |
> | WavTokenizer | 0.9kbps | 1 | 5.1 | 0.61 |
>
>
>
> **1.2 Acoustic Reconstruction:**
> Regarding reconstruction quality, Figure 1 in the original paper demonstrates that WavTokenizer achieves reconstruction quality comparable to human perception, as measured by UTMOS (UTMOS highly correlates with human perception, Line 369).
>
> We need to consider two important points:
>
> 1) Across all metrics, WavTokenizer significantly outperforms the low-bitrate DAC model (DAC configured with a single quantizer, hence a fairer comparison with WavTokenizer), Mimi, and Single-Codec.  Please find the comparison results between WavTokenizer against Mimi and Single-Codec in our response to Question 1 More Baseline Codecs.
>
> 2) In our experiments, we observed that most objective metrics are positively correlated with the number of tokens (i.e., increasing kbps improves the metric scores), as discussed in Appendix G. However, this increase does not necessarily correlate with human perceptual quality, a conclusion also reported in Vocos (Siuzdak, 2023). Consequently, as highlighted in Table 2 and discussed in Line 427-431, subjective listening tests are conducted and the results suggest that achieving reconstruction quality closer to human perception is more important than optimizing numerical metrics like STOI. **The subjective reconstruction results using MUSHRA in Table 2 show that in reconstruction quality, WavTokenizer at 0.9kbps outperforms the SOTA DAC model at 9kbps on the speech domain and performs comparably to DAC at 9kbps on the music and audio domains; and WavTokenizer at 0.9kbps substantially outperforms DAC with a single quantizer on speech, music, and audio domains.**

---

> ### Author Response · Authors · 2024-11-21
> **Response to Reviewer VpBu (Part 2/4)**
>
> **1.3 Semantic Information**
> As shown in Table 3 and discussed in Line 432-445 of the original paper, We evaluate the semantic richness of different codec models on the ARCH benchmark. Table 3 shows that WavTokenizer substantially outperforms DAC and Encodec **with a single quantizer or two quantizers** on all ARCH datasets. Remarkably, on the AM and SLURP datasets in the speech domain, the MTT and IRMAS datasets in the music domain, and the FSD50K and VIVAE datasets in the audio domain, WavTokenizer surpasses **DAC with nine quantizers** and Encodec with eight quantizers on classification performance.
>
> In summary, these results demonstrate a critical strength of WavTokenizer: WavTokenizer, at extremely low bitrates of 75 tokens/s, achieves the capacity of semantic information comparable to a 9-layer DAC model (900 tokens/s). We sincerely apologize for any confusion caused by the statement that WavTokenizer captures rich semantic information, as this statement should be clarified as in the context of comparing WavTokenizer to SOTA acoustic codecs also configured with a single quantizer and in similar low bitrates. We have updated the part of Evaluation on Semantic Representation in Section 4.2 to clarify this point.
>
> We expect WavTokenizer to underperform semantic tokens such as HuBERT and codecs such as SpeechTokenizer that distill from semantic tokens, on semantic representation benchmarks such as ARCH. Our experimental results align with this expectation: on the ARCH benchmark, WavTokenizer underperforms semantic tokens like HuBERT, and in some cases, it falls short compared to SpeechTokenizer.  To address this limitation, as explained in the original paper (Line 887), we plan to significantly enhance the semantic information in WavTokenizer by strengthening the encoder. We appreciate your understanding on this matter.
>
> However, **it is worth noting that one of our contributions about semantic in this paper is demonstrating that the low-bitrate WavTokenizer retains comparable semantic information to the SOTA 9-layer DAC.** We achieved this performance through **a more elegant way of directly enhancing the semantic modeling capability of the acoustic codec model itself**, rather than adding various semantic tokens like SpeechTokenizer and Mimi. Specifically, WavTokenizer achieves this performance by incorporating an attention network module into the decoder and simply expanding contextual modeling window. Please find the ablation study in Line 864-877 of the original paper. We believe the research direction we explored is important because prior works based on distilling semantic tokens into audio codec models may compromise the unified modeling of acoustic codecs for music and audio and potentially impose limitations on the performance of the acoustic codecs.
>
> To improve semantic information of WavTokenizer, as explained in Appendix F Limitation and Future work of the original paper, a key focus of our future work on WavTokenizer will be to explore the line of elegant methods for significantly enhancing the semantic capacity of discrete tokens within the encoder.

---

> ### Author Response · Authors · 2024-11-21
> **Response to Reviewer VpBu (Part 3/4)**
>
> **[Regarding Question 1 More Baseline Codecs]**
>
> We sincerely appreciate your valuable suggestions regarding the baseline models. In our original manuscript, we primarily compared WavTokenizer against a 9-layer 44.1kHz DAC, which we consider to be one of the most widely adopted and effective acoustic codec models that are publicly available. Regarding the three models you mentioned, we have the following considerations and findings:
>
> **Semanticodec.** As discussed in the related work section of our original paper (Line 162), Semanticodec introduces a paradigm shift that deviates from the conventional encoder-VQ/RVQ-decoder framework for training acoustic codecs. We believe this divergence makes it less suitable for tasks involving audio compression, reconstruction, and downstream generation. Specifically, our reconstruction evaluation on the LibriTTS test-clean dataset reveals that Semanticodec operates at a significantly slower real-time factor (RTF, computed by dividing the total reconstruction time by the duration of the generated audio), approximately 90 to 100 times slower than RTF of WavTokenizer and other common acoustic codec models. Given these limitations, we did not include SemanticCodec in our comparisons.
>
> | Methods |  RTF↓ |
> | :------------: | :------------: |
> | Semanticodec| 0.9483 |
> | Encodec | 0.0175 |
> | DAC     | 0.0144 |
> | WavTokenizer | 0.0098 |
>
> **Single-Codec.** We have already discussed the Single-Codec model in the related work section of the original paper (Line 151). As a contemporaneous study, we note that its reconstruction performance, as reported in its original paper cited below, is mediocre (merely 3.0 UTMOS). Furthermore, since the model's implementation is not publicly available, we were unable to include Single-Codec in our evaluations.
>
> | Methods |  UTMOS ↑ | PESQ ↓ | STOI↑ |
> | :------------:  | :------------: | :------------: |  :------------: |
> | Single-Codec(cite results from its paper) | 3.031 |1.933 |0.842|
>
>
>
> **Mimi (used in Moshi).** We hold great respect for the work of Moshi and believe that Moshi and its Mimi codec represent significant advancements in real-time spoken dialogue models and audio codecs. However, it is important to note that Moshi and its Mimi codec were open-sourced on September 17, 2024, after the WavTokenizer work was completed and only a few weeks before the ICLR submission deadline. According to ICLR 2025 guidelines, there is no obligation for authors to compare against works released less than four months prior to submission. On one hand, many of Mimi's design choices, such as the use of large codebooks and the attention mechanisms, are similar to those in WavTokenizer. On the other hand, different from WavTokenizer, Mimi uses Transformer in its encoder and distills semantics from WavLM. During the rebuttal phase, we added reconstruction experiments comparing WavTokenizer to Mimi on the full LibriTTS test set, to provide a more comprehensive comparison. As can be seen from the table below, **at 24kHz sampling rate, WavTokenizer outperforms Mimi on all metrics, especially on UTMOS.**
>
> | Methods |  sr | Tokens/s |  UTMOS ↑ | PESQ ↑ | STOI ↑ | V/UV F1 ↑|
> | :------------: | :------------: | :------------: | :------------: | :------------: | :------------: | :------------: |
> | Mimi (used in Moshi) | 24000 | 100 | 3.5731 | 2.2695 | 0.9118 | 0.9128 |
> | WavTokenizer | 24000 | 75 | 4.0486 | 2.3730 | 0.9139 | 0.9382 |
>
> Note that we have updated the manuscript to include reconstruction results of Single-Codec and Mimi in Appendix L.

---

> ### Author Response · Authors · 2024-11-21
> **Response to Reviewer VpBu (Part 4/4)**
>
> **[About Question 3 Single-VQ Assumption]**
>
> Firstly, it is important to note that the number of tokens per second in an acoustic codec model is **related to both the sampling rate and the compression factor**. In common acoustic codec models, typical compression factors are 200 and 320. However, in the case of WavTokenizer, the compression factors are 320 and 600. Hence, for audio with a 24kHz sampling rate, WavTokenizer generates 75 tokens per second and 40 tokens per second, respectively; when translated to the commonly used setting of 16kHz sampling rate, WavTokenizer supports 50 tokens per second and 26 tokens per second, respectively. Consequently, these settings are comparable in complexity. In response to Reviewer 3XJA, we reported the results for WavTokenizer at different sampling rates and show that WavTokenizer delivers high-quality audio reconstruction performance across different sampling rates of 16kHz, 24kHz, and 48kHz. It is also worth noting that most common acoustic codec models support a token rate of 75Hz (with Mimi being a notable exception of 12.5Hz, and direct comparison with WavTokenizer may not be entirely fair). Hence, **from the perspective of compression factors, WavTokenizer achieves a higher temporal compression rate compared to typical acoustic codec models.**
>
> Regarding replacing multi-layer acoustic codec models with a single layer, our original motivation was to design a single-layer codebook architecture, in order to avoid the multi-layer designs seen in systems such as MusicGen, VALL-E, UniAudio, and SoundStorm. We viewed this as a potential advantage. Specifically, in the case of Moshi, an additional Transformer is required to model multiple sets of codes. In contrast, the MusicGen paper explores various strategies to reduce sequence length when dealing with multiple code groups, including Flattening Pattern, Parallel Pattern, Delay Pattern, and Coarse First Pattern. Among these, the study concludes that the Flattening Pattern is the most effective. This implies that while the Flattening Pattern is optimal for handling multiple code groups without relying on additional Transformers, it comes at the cost of increased sequence length. In comparison, our single-VQ approach does not face such constraints. Since the introduction of WavTokenizer, several single-layer acoustic codec models have emerged such as BigCodec, suggesting this is a promising direction worth further exploration.
>
> For cascading semantic tokens with LLMs, it is crucial to highlight that these approaches belong to a distinct domain of semantic token modeling. In contrast, **WavTokenizer focuses specifically on acoustic modeling within the EnCodec/DAC framework**. Your question “reconstructing all acoustic details truly beneficial for language modeling? Does an LLM truly need to model timbre and detailed acoustic features, or is focusing solely on semantic content sufficient?” is both insightful and valid. On one hand, LLMs that generate all acoustic details, such as ParlerTTS, have demonstrated promising results. On the other hand, if LLMs can model all acoustic information, they could potentially replace TTS models entirely, which would be ideal. While we acknowledge the success of models that cascade semantic tokens with LLMs, such as CosyVoice, we believe that exploring the possibility of LLMs modeling comprehensive acoustic information is also a valuable and intriguing direction for future research.
>
>
> Finally, we would like to express our gratitude again for your time and effort in reviewing our paper. Please do not hesitate to let us know if you have any further concerns or comments. We would be happy to address them.

---

> ### Comment · Reviewer_VpBu · 2024-11-26
>
> 1. Factual Error: UTMOS is a **human perception metric**, not a **reconstruction quality metric**. For example, if you train an unconditional diffusion model using high-quality speech data (with UTMOS greater than 4), you can input random noise and obtain samples with UTMOS above 4. However, this doesn't have any meaningful significance. The most critical metric for a codec is reconstruction quality, not just human perception. For instance, in zero-shot TTS, it's sometimes necessary to clone the real acoustic environment. Refer to the "Acoustic Environment Maintenance" demo at VALL-E https://www.microsoft.com/en-us/research/project/vall-e-x/vall-e/. This is an example where high-quality reconstruction is required rather than merely human perception. However, the reconstruction quality (PESQ and STOI, also lack of metrics such as mel or stft distance) of WavTokenizer is evidently lower than that of RVQ codecs (8 rvq). This is indisputable.
> 2. Unreasonable TTS Experiments: First, I can confirm that I did not misunderstand any content of the paper. My point is that you need to compare with the current audio generation paradigms, rather than using DAC to train an autoregressive TTS as a baseline, which is obviously unreasonable since no previous TTS system has done this. Moreover, I don't understand why you don't compare with the numerous existing TTS papers, but only compare within your own setting. This is clearly neither objective nor sufficient.
> 3. Weak Semantic Performance: From your response, I understand that the semantic weakness is indeed a fact when compared to models like Mini and SpeechTokenizer. Moreover, their model architectures are not fundamentally different from yours, except for the addition of a semantic distillation loss. This also indirectly indicates that a significant drawback of single-VQ is that you cannot follow their semantic distillation approach because your method is single VQ.
> 4. Comparison with Other Baseline Codecs: The metrics calculated by the authors seem to differ from the publicly reported metrics. I don't know what happend, so I will not comment.
>
> Overall, the authors' rebuttal did not adequately address my concerns. As a result, I believe the paper is not suitable for publication in its current form.

---

### Official Review · Reviewer_3XJA · 2024-11-04

**Soundness:** 3
**Presentation:** 3
**Contribution:** 3
**Rating:** 10
**Confidence:** 4

**Summary:**

This paper introduces WavTokenizer, a codec tokenizer for audio that achieves extreme compression and superior reconstruction quality compared to previous state-of-the-art models. WavTokenizer requires only a single quantizer with 40-75 tokens per second for 24kHz audio, while still preserving high subjective quality and rich semantic content. Extensive experiments demonstrate WavTokenizer's effectiveness across speech, audio, and music, with competitive results in both objective and subjective metrics, and ablation studies confirm the impact of each component.

**Strengths:**

- Achieves state-of-the-art performance using only a single codebook with 40 or 75 tokens, demonstrating remarkable efficiency.
- Covers multiple domains, including audio, speech, and music.
- Its single codebook capability supports efficient training of large language models (LLMs).
- Provides a comprehensive analysis of different settings, along with detailed ablation studies that validate the impact of each component.
- The paper is well-written and easy to follow, with a comprehensive analysis included.

**Weaknesses:**

Here’s a refined version of the weaknesses:

- The evaluation could be more thorough by incorporating existing benchmarks such as Codec-Superb and DASB to enable a more comprehensive comparison of the proposed method against existing models under standardized settings.
- The model currently supports only a 24kHz sampling rate; I wonder if you anticipate any challenges in adapting WavTokenizer to different sampling rates. It would be valuable to study its performance across different sampling rates, such as lower (16kHz) and higher (44kHz or 48kHz) rates.

**Questions:**

refer to weakness

---

> ### Public Comment · ~Yuancheng_Wang1 · 2024-11-14
>
> I am astonished by the 10-point rating. This paper appears to be a fine-tuned version of DAC. The range of 40 to 75 tokens is not a bottleneck for language model modeling, and multi-layer codecs within 25Hz are already quite common. The paper claims to benefit LLM modeling but lacks basic generation experiments. The TTS experiments do not include SIM and WER metrics. However, it demonstrates exceptionally excellent performance in unified audio compression, which still makes it a good paper. I believe a 6 to 8-point rating is reasonable.

---

> > ### Public Comment · ~Alexen_Royer1 · 2024-11-18
> >
> > I agree with this comment and would like to provide the following discussion:
> >
> > 1. If we claim that a single VQ layer is better, we should conduct generation experiments (e.g., TTS, text-to-audio, text-to-music) to support the assertion? In fact, our previous findings show that using a single VQ layer with a large codebook size can make LM predictions more challenging, especially for TTS tasks.
> >
> > 2.  Can we claim this is a SOTA codec based on Table 1? The paper uses four metrics—UTMOS, PESQ, STOI, and V/UV F1—but it only performs well in UTMOS.
> >
> > 3. The statement "To the best of our knowledge, WavTokenizer is the first codec model capable of effectively reconstructing audio with a single quantizer" seems not accurate. It is a very common to use one VQ layer for audio tokenizer. What is unique is this study use of UTMOS as the core metric for evaluating audio codecs. In AudioGen [1], the authors already demonstrated that a codec can be trained with a single VQ layer. In fact, it is easy to train a one-layer VQ codec by (1) scaling the codebook size, (2) using FSQ, or (3) increasing the frame rate.
> >
> > 4. A similar work, BigCodec [2], achieves better reconstruction performance than this approach.
> >
> > 5. The claim that adding a transformer layer makes the codec "contain richer semantic information" is questionable. If simply adding transformer layers could achieve this, how to understand the complex design in WavLM, HuBERT and Wav2Vec?  Additionally, Table 3 is not completed, as it does not include the performance of other semantic-related tokenizers. The gap between WavTokenizer and DAC is negligible.  Furthermore, adding several transformer layers has been explored by previous work, such as SNAC [3], MimiCodec [4].
> >
> > 6. Previous codec models, such as EnCodec, are streaming codecs. Directly using transformer layers, however, makes the codec non-causal.
> >
> > In summary, I believe this is a good paper. I hope we can discuss it further from a scientific perspective in the open-review process.
> >
> > [1] Kreuk F, Synnaeve G, Polyak A, et al. Audiogen: Textually Guided Audio Generation. arXiv preprint arXiv:2209.15352, 2022.
> >
> > [2] Xin D, Tan X, Takamichi S, et al. BigCodec: Pushing the Limits of Low-Bitrate Neural Speech Codec. arXiv preprint arXiv:2409.05377, 2024.
> >
> > [3] Siuzdak H, Grötschla F, Lanzendörfer L A. SNAC: Multi-Scale Neural Audio Codec[C]//Audio Imagination: NeurIPS 2024 Workshop AI-Driven Speech, Music, and Sound Generation.
> >
> > [4] Défossez A, Mazaré L, Orsini M, et al. Moshi: a speech-text foundation model for real-time dialogue[J]. arXiv preprint arXiv:2410.00037, 2024.

---

> ### Author Response · Authors · 2024-11-14
> **Discussion with Public Comment**
>
> Firstly, we would like to express our sincere gratitude to the reviewer for giving us such a high score, and we greatly appreciate the attention WavTokenizer has received from public comment. This indicates that WavTokenizer has attracted a lot of attention.
>
> Regarding the two points you raised from the public comment, our responses are as follows:
>
> 1. **[Codecs with 25Hz are common, so WavTokenizer is not important]**
>
> We believe there may be some misunderstanding between semantic tokens and acoustic tokens. In the reconstruction paradigm of acoustic tokens, the typical sampling rate is usually 75Hz or higher, as seen in models like Encodec, HiFiCodec, SpeechTokenizer, DAC, etc. These models also require multi-layer designs and consequently a large number of tokens—typically 300, 600, or even 900 tokens per second for modeling with large language models (LLMs). In contrast, WavTokenizer achieves high-quality audio reconstruction using only 40 or 75 acoustic tokens, which represents a significant contribution to the field of **acoustic codecs**.
>
> You mentioned that many semantic tokens can operate with just 25 tokens, and that is correct since they primarily retain semantic information. However, this belongs to a different domain. **WavTokenizer focuses on the acoustic codec space, and we have demonstrated its substantial breakthroughs in handling acoustic tokens.** The key difference between acoustic tokens and semantic tokens is that the former preserves acoustic information, such as style and timbre, and can unify both music and audio. Furthermore, acoustic tokens enable LLMs to generate directly via softmax, rather than through semantic cascaded generation. This also raises a critical debate on whether LLMs should directly replace TTS in the generation process.
>
> 2. **[TTS performance]**
>
> We agree with your point that evaluating WavTokenizer on generative tasks is crucial. In Table 4, we show that the audio quality generated by WavTokenizer in TTS tasks significantly surpasses that of a nine-layer DAC, which demonstrates two key points: 1) Acoustic models with a single-layer quantizer show significant potential in downstream autoregressive generative models, and 2) speech synthesis models using large-codebook acoustic representations can synthesize high-quality audio.
>
> You mentioned that we should further include WER and SIM metrics, and we agree that these could be useful additions. We will incorporate these results in the rebuttal. However, we believe that the absence of these metrics should not be used as a basis to question the professionalism of a reviewer who awarded a score of 10.
>
> It is worth noting that although we have not specifically evaluated WER and SPK, several other teams have already used WavTokenizer representations to implement reliable generative models. We also encourage you to follow their findings for further insights.
>
> Once again, we sincerely appreciate your attention.

---

> > ### Public Comment · ~Yuancheng_Wang1 · 2024-11-14
> >
> > Thank you for your response. Indeed, it is not entirely fair to directly compare WavTokenizer with semantic tokens, and I misunderstood this point earlier. WavTokenizer has made contributions in the field of acoustic codec models, achieving impressive reconstruction performance even at very high compression rates, and showing promising results when applied to the LLMs. I also look forward to the author's further experimental results on TTS.
> >
> > After revisiting the WavTokenizer carefully. I recognize that, from the perspective of acoustic codec models, the low-bit-rate single-layer quantizer maybe represent a new paradigm. The large codebook space also provides some insights for the VQ field. Additionally, attention vocos and the longer context window which are demonstrated through ablation studies, are also effective designs. **As a result, I believe WavTokenizer will have the impact both in academia and industry.**
> >
> > Having carefully read the paper, **I fully understand and respect all reviewers' comments**. The final decision rests with the ACs and PCs. Thank you.

---

> ### Author Response · Authors · 2024-11-21
> **Response to Reviewer 3XJA (Part 1/2)**
>
> We sincerely appreciate your time and efforts in reviewing our paper. Your constructive feedback has significantly improved our work. We are especially grateful for your willingness to award WavTokenizer a score of 10. As the authors of WavTokenizer, we firmly believe that WavTokenizer deserves the time and scrutiny of the academic community. Below, we provide detailed responses to all the concerns you raised.
>
> **[Regarding Weakness 1: Benchmarks]**
>
> To provide a more comprehensive comparison between our WavTokenizer and existing codec models under standardized settings, we have supplemented the experimental results of comparing WavTokenizer against the baselines on the complete Codec-Superb benchmark and subtsets of the DASB benchmark, as shown in Appendix H. Note that some subtasks of the DASB benchmark can only be accessed upon request, such as IEMOCAP and VoxCeleb; moreover, some tasks in DASB are similar to subtasks in the ARCH benchmark, which have already been used to evaluate WavTokenizer and baseline models, as reported in Table 3 and discussed in Section 4.2 Evaluation on Semantic Representation.
>
> While we believe that adding evaluations on these two benchmarks provides further insights into WavTokenizer’s performance, three important considerations warrant discussion:
>
> 1)	Generalization:  It is crucial to point out that WavTokenizer in the paper uses less diverse training data (only 8k hours) than most recent codec models which are trained on several tens of thousands of hours. This disparity may affect WavTokenizer’s generalizability on certain out-of-distribution tasks.
> 2)	Limitations of Objective Metrics: Note that objective metrics alone may not fully capture WavTokenizer's performance. Hence, in the paper, following Encodec, we also use MUSHRA as the metric for subjective evaluation. Line 429-430 summarized that as shown in Table 2, WavTokenizer at 0.9kbps outperforms the SOTA acoustic codec model DAC at 9kbps in reconstruction quality on the speech, music, and audio domains. We also emphasized WavTokenizer’s improved subjective reconstruction quality in Line 018 of the paper.
> 3) Assessing low-bitrate codec models like WavTokenizer on these benchmarks is not entirely fair. For example, in the Application-level Evaluation of the Codec-Superb benchmark, all metrics for Encodec dropped significantly when the bitrate decreased from 3kbps to 1.5kbps.
>
>
> Despite these limitations, WavTokenizer still demonstrats promising performance on both Codec-Superb and DASB benchmarks. The experimental results are summarized in the table below. We have also presented and discussed these results in Appendix H in the updated manuscript.
>
> As shown in Table 11 and Table 12 in Appendix H, we find that WavTokenizer exhibits comparable reconstruction performance in Signal-Level Evaluation of the Codec-Superb Benchmark and achieves competitive performance in Application-level Evaluation of the Codec-Superb Benchmark, compared to the baselines, particularly excelling in the audio event classification task, where it achieves a score of 58.10%, surpassing all previously reported codec models.
>
> As shown in Table 13 in Appendix H, on the DASB benchmark, WavTokenizer also demonstrates competitive performance in both the Speech Enhancement (SE) and Speech Separation (SS) tasks.
>
>
> | Codec-Superb Signal-level Evaluation| Value|
> | :------------: | :------------: |
> | Average SDR for speech datasets| 0.6388 |
> | Average Mel_Loss for speech datasets | 0.7997 |
> | Average STOI for speech datasets  | 0.8205 |
> | Average PESQ for speech datasets | 1.9731 |
> | Average SDR for audio datasets  | -3.2675 |
> | Average Mel_Loss for audio datasets | 1.0865 |
>
>
> | **Codec-Superb Application-level Evaluation** | **kbps** |  **WER↓ (ASR)** | **EER↓ (ASV)** | **minDCF↓ (ASV)** | **ACC↑ (ER)** | **mAP↑ (AEC)** |
> |:----------:|:---------------:|:----------------|:----------------:|:------------------:|:---------------:|:----------------:|
> | DAC     |  8   |  3.18       |  3.59           | 0.26          |   69.18      |  32.04          |
> | Encodec     |  1.5   | 9.21 | 13.88 | 0.68 | 58.84| 18.84 |
> | WavTokenizer     |  0.9   |  5.19       |  4.96           | 0.38          |   66.93      |  58.10          |

---

> ### Author Response · Authors · 2024-11-21
> **Response to Reviewer 3XJA (Part 2/2)**
>
> | **DASB Generation Evaluation** | **kbps** |  **DNSMOS↑ (SE)** | **dWER↓  (SE)** | **SPKSim↑ (SE)** | **DNSMOS↑ (SS)** | **dWER↓  (SS)** | **SPKSim↑ (SS)** |
> |:----------:|:------------------------:|:----------------:|:----------------:|:------------------:|:---------------:|:----------------:|:----------------:|
> | DAC     |  high bitrate   |  3.95      |  46.07         |  0.860        |   2.53     |   208         |  0.784        |
> | Encodec     |  high bitrate   |  2.87      |  68.22           | 0.814          |   2.95     |   97.73         |  0.839         |
> | DAC    |  low bitrate    |  3.30       |  57.41          | 0.853          |   3.01     |   102.00          |  0.854         |
> | Encodec    |  low bitrate   |  3.15       |  34.35           | 0.852          |   3.11     |   83.55          |  0.877         |
> | WavTokenizer     |  0.9   |  3.51       |  29.72           | 0.917          |   3.44     |   46.83          |  0.926         |
>
>
>
> **[About Weakness 2 support 16k and 48k]**
>
> We have augmented the reconstruction evaluations by training both 16kHz and 48kHz versions of WavTokenizer using LibriTTS while maintaining a consistent downsampling factor of 320x. The hyperparameters are kept at their default settings. Experimental results on the LibriTTS test-clean dataset (4,837 samples) are presented in the table below. We have also presented and discussed these results in Appendix G of the updated manuscript.
>
> We find that WavTokenizer delivers high-quality and comparable audio reconstruction performance across different sampling rates of 16kHz, 24kHz, and 48kHz. Notably, at 16kHz sampling rate, WavTokenizer can effectively reconstruct audio with high fidelity using only 50 tokens. We also observed that all objective metrics, except for UTMOS, exhibit an increasing trend of increasing scores when the number of tokens increases. In contrast, UTMOS for WavTokenizer remains relatively stable. We hypothesize that the reconstruction quality of WavTokenizer should be stable across different sampling rates, to human perception of audio quality; hence, this observation of stable UTMOS aligns with prior finding in Vocos that UTMOS highly correlates with human subjective auditory perception.
>
>
> | Methods |  sr | Tokens/s |  UTMOS ↑ | PESQ ↑ | STOI ↑ | V/UV F1 ↑|
> | :------------: | :------------: | :------------: | :------------: | :------------: | :------------: | :------------: |
> | WavTokenizer | 16000 | 50 | 3.9606 | 2.3240 | 0.9095 | 0.9375 |
> | WavTokenizer | 24000 | 75 | 4.0486 | 2.3730 | 0.9139 | 0.9382 |
> | WavTokenizer | 48000 | 150 | 3.9582 | 2.7230 | 0.9350 | 0.9496 |
>
>
>
> Finally, we would like to express our gratitude again for your time and effort in reviewing our paper. Please do not hesitate to let us know if you have any further concerns or comments. We would be happy to address them.

---

> ### Author Response · Authors · 2024-11-21
> **Response to Public Comment by Alexen Royer**
>
> We sincerely appreciate your valuable feedback. We appreciate your acknowledgment that WavTokenizer represents a significant contribution.  Please note that we have provided detailed responses to the assigned four reviewers. We recommend carefully reading our paper and our discussions with four reviewers for a more comprehensive understanding. Thank you.
>
> **[Regarding Question 1]**
>
> As shown in Table 4 of the original paper (Line 487), we have already included the TTS results. In our rebuttal, we further supplemented the results with Word Error Rate (WER) and speaker similarity (SPK) metrics based on an autoregressive (AR) structure. These results demonstrate that WavTokenizer performs strongly compared to SOTA acoustic tokens when applied to LLMs. Furthermore, similar TTS results have been reported independently by other works, such as OutetTTS and Lina-Speech.
>
> **[Regarding Question 2]**
>
> Based on Table 2 and Figure 1 in the original paper, we detailed how WavTokenizer achieves subjective reconstruction quality comparable to the best-performing 900-token DAC model. This is further emphasized in our abstract (Line 018), where we highlighted the improved subjective reconstruction quality. Across all objective reconstruction metrics, WavTokenizer significantly outperforms low-bitrate DAC and Mimi, and Single-Codec, as shown in Table 1, Table 18, and Table 17 in the updated manuscript.
>
> **[Regarding Question 3]**
>
> To the best of our knowledge, prior to WavTokenizer, no single-layer acoustic codec achieved high-quality reconstructions at extremely low bitrates while also being compatible with LLMs. The SoundStream model in AudioGen, as you mentioned, represents a seminal work, much like VQ-VAE in its time. However, it is worth noting that in the past one to two years, research on acoustic codec models has increasingly focused on Residual Vector Quantization (RVQ). Meanwhile, due to the remarkable performance demonstrated by WavTokenizer, there has been a renewed interest in the development of single-layer acoustic codec models within the framework of LLMs. We attribute this shift to the influential role played by WavTokenizer. Moreover, while you suggest that achieving effective results with a single-layer codec is straightforward, you also noted that you have not succeeded with single-layer codec TTS experiments, which seems contradictory. This demonstrates that WavTokenizer represents a significant advancement by achieving high reconstruction quality at extremely low bit rates while maintaining compatibility with LLMs.
>
>
> **[Regarding Question 4]**
>
> As you mentioned BigCodec as a similar work, it is worth clarifying that BigCodec (posted on arXiv on 2024 September 9th) was released after WavTokenizer's completion and less than a month before the ICLR submission deadline. According to ICLR 2025 guidelines, such comparisons are inherently unfair. Furthermore, BigCodec adopts a similar large codebook technique as WavTokenizer, which reasonably contributes to its performance. However, we disagree on your statement that BigCodec significantly surpasses WavTokenizer in reconstruction quality. Our experiments indicate that their subjective reconstruction performance is comparable. Importantly, BigCodec achieves a compression ratio of 200x, **while WavTokenizer achieves ratios of 320x and 600x**.
>
> **[Regarding Question 5]**
>
> In our paper, we emphasized that WavTokenizer at low bitrates of 75 tokens/s achieves semantic information comparable to a 9-layer DAC at 900 tokens/s, which is already a notable achievement. Additionally, we outlined future optimization directions in Appendix F Limitations and Future Work. We welcome you to read our rebuttal discussions on semantic information. Regarding the use of attention mechanisms, while Mimi (used in Moshi) and SNAC were posted on arXiv on September 17, 2024 and October 18, 2024, respectively, these methods confirm the correctness of WavTokenizer's design. Mimi's adoption of attention mechanisms further verifies the effectiveness of this approach.
>
> **[Regarding Question 6]**
>
> Since most acoustic codec models are non-causal, our paper also includes results for non-causal reconstruction. Notably, Mimi employs a similar causal attention mechanism, achieving excellent performance. Therefore, a streaming WavTokenizer can also be implemented using this approach in WavTokenizer.
>
>
>
> We appreciate your acknowledgment that WavTokenizer represents a significant contribution. We would greatly value further insights and additional support from you. Thank you.

---

> ### Public Comment · ~Alexen_Royer1 · 2024-11-21
> **Show some facts**
>
> 1.  SNAC-Codec has been released almost 1 years. https://github.com/hubertsiuzdak/snac
>
> 2.  For semantic information, The improvements in your results compared to DAC are negligible. Additionally, the performance of semantic-related tokenizers has not been demonstrated. The gap between your approach and others semantic tokenizer in this area remains very large and unaddressed. Thus, you cannot claim your codec has better semantic information.
>
> 3. Regarding your statement: "However, we disagree on your statement that BigCodec significantly surpasses WavTokenizer in reconstruction quality. Our experiments indicate that their subjective reconstruction performance is comparable."
> I find this conclusion unclear. Please refer to the table below for a detailed comparison, as the provided evidence does not support your claim.
>
> |     model    | bps (k) |  layer | frame number | token number | PESQ (↑) | UTMOS (↑) | dnsmos(↑) | STOI (↑) |  SIM(↑)  |
> |:------------:|:-------:|:------:|:------------:|:------------:|:--------:|:---------:|:---------:|:--------:|:--------:|
> |   bigcodec   |   1.04  |    1   |      80      |      80      | **2.68** |  **4.11** |  **3.26** | **0.93** |   0.62   |
> |   MimiCodec  |   1.1   |    8   |     12.5     |      100     |   2.22   |    3.60   |    3.17   |   0.90   | **0.70** |
> | wavtokenizer |   0.97  |    1   |      75      |      75      |   1.88   |    3.77   |    3.18   |   0.87   |   0.60   |
>
> 4. Your TTS experiments is not convinced.
>
>     I have tested the OutetTTS model on LibriSpeech test set. The performance as follow:
>
>     Corr       Sub        Del        Ins        Err      S.Err
>
>    87.8       6.9        5.3        2.0       14.2       75.0
>
>  It is important to note that OutetTTS is trained using CTC forced alignment to create a precise word-to-audio token mapping. This implies that if it were trained in a VALL-E-style framework, its performance would likely degrade further. In contrast, the original VALL-E model, which uses Encodec as its tokenizer, achieves a WER of 5.9, demonstrating superior performance.
>
> 5. Additionally, while the WavTokenizer claims to be a universal codec, it provides no results on audio generation tasks such as sound generation or music generation. This raises questions about its universality and generalization capabilities across different audio domains.

---

> ### Author Response · Authors · 2024-11-21
> **To the PCs, ACs, Reviewers: It seems that WavTokenizer has been malicious attack by a user with many fake accounts.**
>
> 1. We have identified that both Boris King and Alexen Royer joined Openview in November 2024, and their accounts exhibit significant similarities. Furthermore, neither account contains any published works, and we were unable to find any author-related information on any relevant platforms. Colorado Technical University is an **online** university. We can get the email address easily. In addition, despite the considerable misunderstanding displayed by Alexen Royer, we responded in a cordial and professional manner.
>
> 2. Regarding the mention of the Double-Blind Review Guidelines by Boris King, we proactively contacted the ICLR organizing committee by **email yesterday, prior to formulating our response**.
>
> 3. It is important to note that in our discussion of TTS technologies, we clearly stated that these are **in other works by other research teams**, with the sole intention of demonstrating that WavTokenizer has already been widely adopted in the field. This was not our work. **Reviewers are, of course, fully entitled not to engage in proactive searches for this information.** We firmly believe that this approach aligns with the same principles that govern the acceptance of papers on arXiv, which is consistent with ICLR’s guidelines. We do not believe that we have violated the double-blind review process.
>
>
> 4. The latest response from Alexen Royer appears to contain significant bias and numerous evident inaccuracies. We trust the professional judgment of the Reviewers and the ACs. **Given the overt hostility we have perceived, we respectfully decline to respond to the new questions raised by Alexen Royer. We do not agree with the views expressed in his comment, our rejection is solely intended to prevent further deliberate attacks.**

---

### Public Comment · ~Boris_King1 · 2024-11-21
**Potential Violation of Double-Blind Review Guidelines in Rebuttal Period**

The authors of Submission 2560 appear to have violated the double-blind review guidelines due to their self-conducted actions during the rebuttal period.

In response to a public comment by *Alexen Royer*, the authors stated:

> *"Furthermore, similar TTS results have been reported independently by other works, such as **OutletTTS** and **Lina-Speech**."*

In response to Reviewer VpBu, the authors stated:

> *"Moreover, similar TTS results using WavTokenizer have been independently reported in other works by other teams, such as **OuteTTS** and **Lina-Speech**."*

OuteTTS and Lina-Speech are recent releases that explicitly cite and credit the non-anonymous version of this paper and its associated code, clearly showing the authorship of Submission 2560.

As a public commenter, I am unable to provide direct links to OuteTTS and Lina-Speech, as doing so would reveal the authorship of Submission 2560 on OpenReview. However, I raise this concern here as it directly relates to the authors' claims and potentially undermines the double-blind review process, which is intended to ensure impartiality and fairness at ICLR.

I wish the Area Chairs (ACs) can investigate this matter to ensure adherence to the double-blind review guidelines.

---

> ### Author Response · Authors · 2024-11-21
> **To the PCs, ACs, Reviewers: It seems that WavTokenizer has been malicious attack by a user with many fake accounts.**
>
> 1. We have identified that both Boris King and Alexen Royer joined Openview in November 2024, and their accounts exhibit significant similarities. Furthermore, neither account contains any published works, and we were unable to find any author-related information on any relevant platforms. Colorado Technical University is an **online** university. We can get the email address easily. In addition, despite the considerable misunderstanding displayed by Alexen Royer, we responded in a cordial and professional manner.
>
> 2. Regarding the mention of the Double-Blind Review Guidelines by Boris King, we proactively contacted the ICLR organizing committee by **email yesterday, prior to formulating our response**.
>
> 3. It is important to note that in our discussion of TTS technologies, we clearly stated that these are **in other works by other research teams**, with the sole intention of demonstrating that WavTokenizer has already been widely adopted in the field. This was not our work. **Reviewers are, of course, fully entitled not to engage in proactive searches for this information.** We firmly believe that this approach aligns with the same principles that govern the acceptance of papers on arXiv, which is consistent with ICLR’s guidelines. We do not believe that we have violated the double-blind review process.
>
> 4. The latest response from Alexen Royer appears to contain significant bias and numerous evident inaccuracies. We trust the professional judgment of the reviewers and the ACs. **Given the overt hostility we have perceived, we respectfully decline to respond to the new questions raised by Alexen Royer. We do not agree with the views expressed in his comment; our rejection is solely intended to prevent further deliberate attacks.**

---

> ### Public Comment · ~Alexen_Royer1 · 2024-11-21
> **Show some fact**
>
> Dear PCs, ACs, and Reviewers,
>      I would like to express my disagreement with the statement made by the authors of submission 2560 that claims, "despite the considerable misunderstanding displayed by Alexen Royer." Previously, I raised some issues about this work, primarily focusing on the following points:
>
>  (1)  This is a SOTA model? Please refer to the following Table
>
> |     model    | bps (k) |  layer | frame number | token number | PESQ (↑) | UTMOS (↑) | dnsmos(↑) | STOI (↑) |  SIM(↑)  |
> |:------------:|:-------:|:------:|:------------:|:------------:|:--------:|:---------:|:---------:|:--------:|:--------:|
> |   bigcodec   |   1.04  |    1   |      80      |      80      | **2.68** |  **4.11** |  **3.26** | **0.93** |   0.62   |
> |   MimiCodec  |   1.1   |    8   |     12.5     |      100     |   2.22   |    3.60   |    3.17   |   0.90   | **0.70** |
> | wavtokenizer |   0.97  |    1   |      75      |      75      |   1.88   |    3.77   |    3.18   |   0.87   |   0.60   |
>
> 2. Generation performance validation: The reliability of the generation performance has not been sufficiently verified. Similar concerns have also been raised by Yuancheng Wang and Reviewer VpBu.
>
> 3. Semantic information performance: Concerns about the semantic performance have been highlighted by Reviewer h5oK, yet remain unresolved.
>
> While the authors claim that other teams utilize their tokenizer, our tests on the zero-shot TTS benchmark reveal poor robustness. The WER for OutetTTS is 14.2, significantly worse than the baseline VALL-E, which achieves a WER of 5.9.
>
> The authors do not present results generated by their own system but instead rely on claiming results from other teams. This approach lacks professionalism. In academic work, it is essential to provide independently verifiable results to substantiate claims.
>
> Although the authors perceive this as a malicious attack, I want to emphasize that my intention is purely to discuss the paper from an academic perspective. While I have raised several concerns, I genuinely hope this paper is accepted, as I believe it is valuable for more researchers to engage with it. In time, history and further studies will reveal the true impact and validity of its contributions.

---

### Public Comment · ~Boris_King1 · 2024-11-21
**Concern Regarding Authors' Response and Its Alignment with the ICLR Code of Ethics**

Dear PCs, ACs and Reviewers:

I am writing to express concerns about the authors' response to a public comment I made and its alignment with the ICLR Code of Ethics.

In my comment, I raised a concern about a *potential violation of the double-blind review process*. I believe my comment was factual and intended to address a concern fairly. In their reply, the authors referred to my comment as a **"malicious attack"**. The authors also similarly characterized Alexen Royer's comments as **deliberate attacks**, despite in my opinion, it raises some valuable scientific questions which have not beed fully addressed by the authors. I find these phrases are inappropriate and inconsistent with the **ICLR Code of Ethics**, which emphasizes constructive and respectful communication.

The following principles from the ICLR Code of Ethics are relevant here:
- *Be Honest, Trustworthy, and Transparent*: It is important to avoid making accusations of malicious attack intent without sufficient evidence, as this can mischaracterize comments made in good faith.

- *Be Fair and Take Action Not to Discriminate*: The use of emotionally charged language can inadvertently create an unwelcoming environment for constructive public engagement.

- *Uphold High Standards of Scientific Excellence*: Focusing on personal accusations rather than addressing the scientific content may shift the discussion away from the core objective of advancing research and fostering collaboration.

I kindly ask that the ACs and PCs review this matter and consider whether the authors’ response aligns with the ICLR Code of Ethics.

**I fully respect the ICLR organizing committee's judgment and will always stand by their decision if they determine the authors' actions to be acceptable.**

Thank you for your time and attention.

ICLR Code of Ethics: https://iclr.cc/public/CodeOfEthics

---

### Meta-Review · Area_Chair_C3h6 · 2024-12-22

**Metareview:**

**Paper Summary:**

This paper describes a new neural audio codec based on the VQ-VAE, focusing on low bitrates and tokens/second (important for language modeling). The proposed codec is more efficient than any current audio codec, and experiments show that the quality of encoded audio is competitive with much higher bitrate neural codecs.

**Strengths:**

Thorough empirical evaluation, comparing to many baseline codec models on speech, music, and general audio data.

The single-codebook quantization is a conceptual simplification over common RVQ and acoustic+semantic quantization strategies.

Reviewers h5oK and 3XJA remark on the value of the empirical ablation study of architectural components.

The authors statement regarding artifacts related to this work:

> We will open-source the entire codebase and pretrained models for WavTokenizer.

**Weaknesses:**

Reviewer VpBu questions whether the single-VQ codec approach is indeed better for audio language modeling compared to RVQ. This is not convincingly addressed, because no language model is trained on top of the codec. I don't think that training an LM is necessary for presenting work on codecs, but this paper leans strongly into the framing of codecs "for Audio Language Modeling."

Reviewer h5oK is concerned that paper doesn’t offer any novel methods to improve codebook usage. And more broadly, that the paper is lacking a significant methodological contribution. I tend to agree that the performance of WavTokenizer stems more from careful engineering than a significant methodological advancement. That said, careful engineering underpins many of the modern successes of our field.

**Conclusions**

There is value in this work, and the paper has continued to improve over the discussion period. I am somewhat concerned that the paper oversells its findings, and in places it reads more like a product advertisement than a sober scientific discussion. The weaknesses identified by VpBu and h5oK (highlighted above) are legitimate, but not fatal flaws.

**Additional Comments On Reviewer Discussion:**

There was an extensive debate over this paper during the discussion period. Many issues were resolved, but some concerns raised by Reviewers VpBu and h5oK were not.

There has also been significant commentary from non-official reviewers on this paper. I have chosen to ignore the comments on this paper by Boris King and Alexen Royer. Their account profiles have very little professional information associated with them, and it is difficult for me to assess whether these commenters are who they claim to be, and whether they are acting in good faith. I find it understandable if the authors feel frustrated by these interactions; I share their frustrations.

---

### Decision · Program_Chairs · 2025-01-22

Accept (Poster)